# How do different pathways connect the stratospheric polar vortex to its tropospheric precursors?

Raphael H. Köhler[1,*], Ralf Jaiser[1,*], and Dörthe Handorf[1]

[1]Alfred Wegener Institute, Helmholtz Centre for Polar and Marine Research, Potsdam
[*]These authors contributed equally to this work.

**Correspondence:** Raphael Köhler (Raphael.Koehler@awi.de), Ralf Jaiser (Ralf.Jaiser@awi.de)

**Abstract.** Processes involving troposphere–stratosphere coupling have been identified as important contributors to an improved subseasonal to seasonal prediction in mid-latitudes. However, atmosphere models still struggle to accurately predict stratospheric extreme events. Based on a novel approach in this study, we use ERA5 reanalysis data and ensemble simulations with the ICOsahedral Non-hydrostatic atmospheric model (ICON) to investigate tropospheric precursor patterns, localised troposphere–stratosphere coupling mechanisms and the involved timescales of these processes in Northern Hemisphere extended winter. We identify two precursor regions: Mean sea level pressure in the Ural region is negatively correlated to the strength of the stratospheric polar vortex for the following 5–55 days with a maximum at 25–45 days, and the pressure in the extended Aleutian region is positively correlated to the strength of the stratospheric polar vortex the following 10–50 days with a maximum at 20–30 days. A simple precursor index based on the mean pressure difference of these two regions is very strongly linked to the strength of the stratospheric polar vortex in the following month. The pathways connecting these two regions to the strength of the stratospheric polar vortex, however, differ from one another. Whereas a vortex weakening can be connected to prior increased vertical planetary wave forcing due to high-pressure anomalies in the Ural region, the pathway for the extended Aleutian region is less straightforward. A low-pressure anomaly in this region can trigger a Pacific/North American (PNA) related pattern leading to geopotential anomalies of the opposite sign in the mid-troposphere over central North America. This positive geopotential anomaly travels upward and westward in time directly penetrating into the stratosphere and thereby strengthening the stratospheric Aleutian High, a pattern linked to the displacement towards Eurasia and subsequent weakening of the stratospheric polar vortex. Overall, this study emphasises the importance of the time- and zonally-resolved picture for an in-depth understanding of troposphere–stratosphere coupling mechanisms. Additionally, it demonstrates that these coupling mechanisms are realistically reproduced by the global atmosphere model ICON.

## 1  Introduction

Seasonal predictability is based on processes that evolve on slower timescales than the synoptic systems of the troposphere. These processes include slow changes in the lower boundaries, such as soil humidity, snow cover, or ocean temperatures, but also the evolution of the stratospheric polar vortex throughout winter. Based on the interactions between planetary and synoptic scales on timescales of some days to multiple weeks, the coupling between stratosphere and troposphere has been identified as

an important source of predictability for a range of processes on subseasonal to seasonal timescales in mid-latitudes in winter (Domeisen et al., 2020a). However, seasonal prediction can only benefit from the slower evolution of stratospheric processes if we develop an in-depth understanding of the involved coupling mechanisms with the troposphere. Moreover, models used for seasonal prediction need to be able to reproduce these processes.

By using the Northern Annular Mode (NAM) to characterise the polar vortex already Baldwin and Dunkerton (2001) showed
that large variations in the strength of the stratospheric circulation are often followed by anomalous tropospheric weather regimes for up to 60 days. The downward propagation of stratospheric anomalies and the effects on surface patterns have been studied thoroughly since then, in particular, for the most dramatic stratospheric phenomenon, the Sudden Stratospheric Warmings (SSW) (e.g. Charlton-Perez et al., 2018; Afargan-Gerstman and Domeisen, 2020; Baldwin et al., 2021; Butchart, 2022). This downward response is well reproduced in model studies and is known to potentially contribute to surface predictability
(e.g. Scaife et al., 2016; Domeisen et al., 2020a). But how well can stratospheric extreme events be predicted in advance? Karpechko (2018) shows that the ECMWF Extended-Range Forecast System is able to predict SSWs with high probability at lead times of 12–13 days. Yet, some SSWs tend to be more predictable then others (Chwat et al., 2022). Overall, the stratosphere exhibits a slower growth of the signal-to-noise problem as compared to the troposphere, and thus, even allows for predictability beyond 2 weeks (Domeisen et al., 2020b). However, they also demonstrate that stratospheric extreme events themselves tend
to exhibit similar predictability to tropospheric weather and, in particular, SSW events tend to be less predictable.

Based on the early studies of Matsuno (1971) on the role of enhanced tropospheric wave forcing in triggering SSWs, different studies have investigated tropospheric precursor patterns to potentially increase the range of stratospheric predictability. It has been shown that pressure changes in certain regions, when in phase, can constructively interfere with the climatological wave-1 and wave-2 pattern, and thus lead to an overall increase of wave driving (e.g. Garfinkel et al., 2010; Smith and Kushner,
2012). In particular the Ural blocking pattern as part of a wave-1 anomaly has been recognised as a precursor pattern for SSWs (e.g. Garfinkel et al., 2010; Cohen and Jones, 2011). Moreover, this blocking pattern has recently received additional attention, as it plays an important role in dynamically linking Arctic amplification and sea ice loss to changes in mid-latitude circulations patterns via a robust but highly intermittent stratospheric pathway (Jaiser et al., 2016; Hoshi et al., 2019; Cohen et al., 2020; Siew et al., 2020; Jaiser et al., 2023): The strong reduction of Arctic sea ice in autumn and winter, in particular in
the Barents and Kara seas, and the associated heating of the overlying atmosphere favour more frequent blocking-type ridges over northwestern Eurasia in early winter, which facilitate the enhanced propagation of wave energy into the stratosphere, where wave breaking can lead to a disruption of the stratospheric polar vortex (Overland et al., 2016; Crasemann et al., 2017). On the Pacific side, Ineson and Scaife (2009) demonstrate that a deeper Aleutian low positively interferes with and strengthens the stationary wave-1 amplitude. Moreover, Bao et al. (2017) identify the positive phase of the Pacific-North American pattern
(PNA) as a precursor of SSWs by means of constructive interference with the climatological planetary wave-1 pattern. Whereas the Ural blocking plays an important role in Arctic–midlatitude linkages, the Aleutian low and the PNA have been identified as a pathway connecting the tropical phenomenon El Niño Southern Oscillation (ENSO) to the strength of the stratospheric polar vortex in winter (e.g. Ineson and Scaife, 2009; Domeisen et al., 2019).

Kretschmer et al. (2017) show that there is a large potential in identifying precursor mechanisms, as they are able to correctly predict 46% of the extremely weak stratospheric polar vortex states for lead times of 16–30 days using a linear regression prediction model based on causal precursors. However, most previous studies average their signals over fixed lag times. Thus, information on the exact timescales of the coupling mechanisms is lost. Based on this, one objective of this paper is to investigate tropospheric precursors of the stratospheric circulation in winter, including a careful analysis of the involved timescales, patterns, and mechanisms. In contrast to earlier studies, we moreover not only focus on SSWs but on precursor patterns of stratospheric circulation by means of the NAM. Thereby, we evaluate the tropospheric precursor patterns and quantify the involved timescales in Sect. 3.1. Moreover, a strong focus lies on the mechanisms and processes involved in coupling tropospheric pressure anomalies to stratospheric circulation changes. Thus, in Sect. 3.2 we investigate the coupling pathways with a novel time and space-resolved method. By comparing results from reanalysis data to ensemble simulations with the global ICOsahedral Non-hydrostatic atmospheric model (ICON), we furthermore test how these troposphere–stratosphere coupling processes are represented in ICON. The ensemble approach in ICON additionally helps us to better quantify the signal-to-noise ratio.

## 2 Data and methods

### 2.1 Data

In this study, we use daily and monthly mean ERA5 reanalysis data (Hersbach et al., 2020) for the period 1979 to 2021. The reanalysis furthermore serves as a reference for the ensemble simulations with the atmospheric general circulation model ICON (Zängl et al., 2015) version 2.1.0. The model is run with the horizontal resolution R2B5, which corresponds to a grid mesh of approximately 80 km, and with 90 vertical levels up to a height of 75 km. The ICON climatology is created by simulating periods from September to May for the years 1979/80 to 2016/17 (38 years), and each period is simulated by five ensemble members. Thus, yielding a total of 190 nine-month simulations. The ensemble members were generated by shifting the initialisation by $\pm 6$ h and $\pm 12$ h. We prescribe mid-monthly sea surface temperatures and sea ice concentrations produced by the Program for Climate Model Diagnosis and Intercomparison for the Atmosphere Model Intercomparison Project (AMIP) experiments of CMIP6 (Durack and Taylor, 2018). Based on the results of Köhler et al. (2021), we use a tuned setup of the subgrid scale orographic (SSO) drag and the non-orographic gravity wave drag scheme, as in particular the stratospheric polar vortex shows a more realistic behaviour. The ICON simulations we use in this study are well tested and correspond to the ICON$_{gwd-}$ experiment of Köhler et al. (2021).

### 2.2 Methods

The basic mean of quantifying the strength of the stratospheric polar vortex in this study is the Northern annular mode (NAM), which is the dominant pattern of dynamic variability in the extratropical Northern Hemisphere (NH). Based on Baldwin and Thompson (2009), we used an empirical orthogonal function (EOF) analysis of the daily, zonally-averaged geopotential anoma-

lies at each pressure level over the area 20°N to 90°N to calculate the NAM. Geopotential height anomalies were obtained by removing the mean seasonal cycle at each grid point, which is calculated for each data set separately (ERA5 and ICON) by averaging the respective geopotential heights day by day over all years. To ensure equal-area weighting, the data were weighted by the square root of the cosine of latitude before performing the EOF analysis (Baldwin et al., 2009). The standardized corresponding timeseries of the first principal components are the NAM indices. Weak (strong) stratospheric polar vortex events are defined by the 20th (80th) percentile of the NAM at 10 hPa. The statistical significance of composite differences in this study is assessed with a two-sided, non-parametric Wilcoxon-test (Bauer, 1972; Hollander et al., 2013).

A Fast Discrete Fourier Transform is applied to investigate changes in the amplitudes of different planetary wave numbers (k=1-3). The wave amplitudes are calculated for each latitude from the climatological monthly mean 500 hPa geopotential height fields of ERA5 and ICON based on the tools implemented by Campitelli (2021).

The influence of tropospheric anomalies on the stratospheric circulation is characterised by vertical wave propagation following Andrews and Mcintyre (1976). The conventional zonal mean Eliassen-Palm (EP) flux is calculated based on tools implemented by Jucker (2021). We use temperature and all three wind components as input without any pretreatment. The reference vertical temperature gradient is treated with a 21-day rolling mean. From the results, we only use the vertical component of EP flux at 100 hPa averaged over 40°N to 80°N to diagnose waves reaching the stratosphere from the main tropospheric wave source.

To be able to analyse sea level pressure and geopotential data in time and space, we have to reduce the dimensionality. Therefore, we implement meridional averaging typically from 45°N to 80°N. During the testing of our methods, we varied this region over Northern Hemisphere extratropical latitude bands with no contradictions found. This keeps the temporal, longitudinal, and vertical dependence as variable arguments in the subsequently described analytical methods.

A regression analysis is performed between a time-shifted dependent variable and an independent variable that is averaged on 10° longitude bins. The daily values of the independent variable are taken from all days of the given month from all years and all ensemble members if applicable. The corresponding diagram shows the regression coefficient that results from anomalies of the independent variable at a given longitude along the x-axis and the dependent variable shifted by days given along the y-axis relative to the independent variable. Regression coefficients with a p-value above 0.05 are considered non-significant and are displayed as hatched areas provided that the null hypothesis is that the slope is zero, using Wald Test (Wald, 1943) with t-distribution of the test statistic. Areas with p-values below 0.001 are additionally encircled.

This concept of regression analysis is extended into three dimensions. Here, the independent variable is a daily index of a given variable at a given position from a given month. The dependent variable is varied along the x-axis for its longitudinal position and shifted in time relative to the index along the y-axis. Additionally, the z-axis shows the vertical dependency on pressure. Both input data have been detrended prior to analysis. The diagram then shows the location of the 10% largest absolute correlation coefficients between the index and dependent variable. These have been verified to easily exceed a 0.001 significance test as described before in terms of the regression analysis.

## 3 Results

### 3.1 Tropospheric precursor patterns and timescales

The main focus of this section lies in identifying tropospheric precursors of anomalous stratospheric NAM and thereby quantifying the involved timescales. The stratospheric polar vortex exhibits its largest variability in January when on the one hand the daily mean, zonal mean westerlies (in 60°N and at 10 hPa) of an undisturbed vortex can reach up to 70 m/s, and on the other hand can reverse allowing for easterlies of more than 20 m/s during vortex breakdowns. Moreover, more frequent Ural-blockings in December and subsequently a weakened stratospheric vortex in January have been identified as important parts

of the stratospheric pathway linking Arctic amplification to stratospheric circulation (e.g. Jaiser et al., 2016; Crasemann et al., 2017). Based on this, we will start our analysis focusing on stratospheric polar vortex strength in January. However, later in the manuscript, we will show that our key findings stay the same for the months of November, December, February, and March (e.g. Fig. 3).

Independent of the type of definition, weak vortex events tend to have a rather typical pattern of downward propagation,

which was first described by Baldwin and Dunkerton (2001) and is often referred to as "dripping paint" plot. Figures 1 a) and b) recreate these famous time–height cross sections, however, for January weak vortex events only. These events are defined by the 20th percentile of the monthly mean NAM index at 10 hPa in January, yielding 9 events in ERA5 and 40 events in ICON. The larger sample size in ICON results in a smoother image, while the qualitative agreement between ICON and ERA5 is shown (cf. Fig. 1). After the onset of negative NAM signals in the upper stratosphere in late December, the signals propagate

downward and remain present in the lower stratosphere throughout January and February. Some of the weak stratospheric NAM signals also propagate to the troposphere leading to a weakening of the tropospheric NAM. The tropospheric signals are larger towards the surface, which is described as "surface amplification" and is caused by reinforcing tropospheric near-surface processes (Baldwin et al., 2019, 2021). On average, weak vortex events in January are preceded (November) and followed (February and March) by a significantly stronger upper stratospheric polar vortex. ICON is able to reproduce the basic known

features of weak stratospheric vortex events in January, also including a realistic downward propagation as well as the upper stratospheric strengthening before and after the event. Although the negative NAM anomalies overall propagate downwards starting in late December, there are also some indications of tropospheric precursor signals in December in ERA5 and ICON. Even when using a very ordinary composite approach for years with a weak stratospheric vortex in January (cf. Fig. 1 c and d), the known mean sea level pressure (MSLP) precursor patterns become visible in December: A significant high-pressure

anomaly over northwestern Eurasia centered over the Ural mountains, and a low-pressure anomaly over the Northern Pacific, centered around the Aleutian islands and Alaska.

Figure 2 offers a time-resolved view of the connection between tropospheric pressure patterns in December and stratospheric NAM in the following. The meridional mean MSLP from 45°N to 80°N in December (x-axis) is regressed onto the NAM at 10 hPa (y-axis) using daily mean ERA5 reanalysis and ICON ensemble data. Figure 1 shows that December sea level pressure is

a predictor for the NAM at 10 hPa, and once again there are two areas that stand out: An area of significant negative regression centered over the Ural region and an area of positive regression in the sphere of influence of the Aleutian low extending from

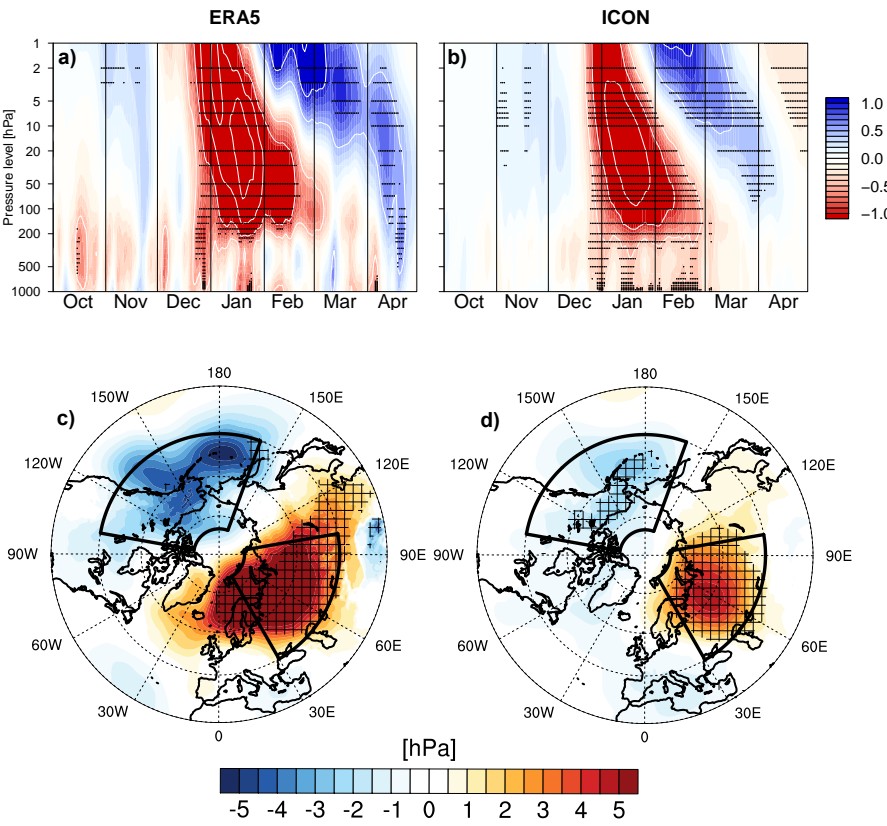

**Figure 1.** Composites of the time–height development of the NAM for weak stratospheric vortex states based on the 20th percentile of the January mean NAM at 10 hPa in ERA5 (a) and ICON (b). The NAM index is nondimensional. The contour interval for the white contours is 0.5. Note that due to common practice negative values (weak NAM) are red and positive values (strong NAM) are blue. Composites of the December mean sea level pressure (MSLP) in hPa for the same January weak vortex events as in (a) and (b) for ERA5 (c) and ICON (d). The Ural region and the extended Aleutian region are marked by black boxes. The definitions of the regions are based on the analysis of Fig. 2. Stippling in a-d indicates statistical significance at the 95% level according to a two-sided Wilcoxon-test.

the Sea of Okhotsk to the Rocky Mountains. The negative regression over the Ural region is strongest at about 60°N and with a lag of 25–45 days, indicating that high (low) pressure in this region is connected to a weakening (strengthening) of the stratospheric polar vortex with approximately one month lag. The maximum of the positive regression is centered over the Rocky Mountains with a lag of 20–30 days, thus indicating that low (high) pressure anomalies in the extended Aleutian low area are connected to a weakening (strengthening) of the stratospheric polar vortex. Due to the large noise introduced by the evaluation of daily data, the maximum correlation coefficients are rather small (0.26 and -0.35 in ERA5, 0.18 and -0.24 in ICON). However, the regression indicates that a reasonable 10 hPa sea level pressure anomaly can push the NAM index by up to 0.5 index points, which will have a significant impact on the stratospheric circulation. The regions of interest are in accordance with the precursor patterns depicted in Fig. 1. Both patterns show significant signals after a few days and last for around 50

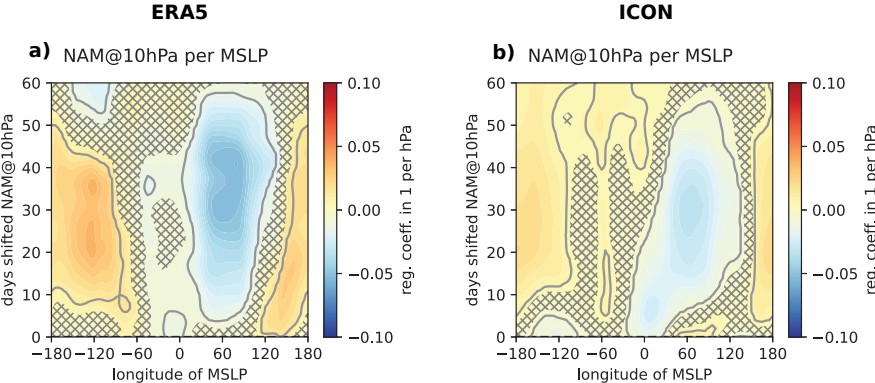

**Figure 2.** Regression of 45°N to 80°N meridional mean MSLP onto the 10 hPa NAM. MSLP is averaged over 10° longitude bins with their respective center positions given along the x-axis. Daily values of MSLP from December of ERA5 data (a) and all years and ensemble members of ICON data (b) are used. Data of the 10 hPa NAM is shifted by days given on the y-axis relative to MSLP data. Areas with regression coefficients with p-values above 0.05 are hatched, and areas with p-values below 0.001 are encircled.

days. ICON reproduces the described patterns, with a maximum negative regression at a lag of 30 days, and a maximum positive regression at 20–30 days lag. However, the maximum of the positive regression is shifted towards the antimeridian and thus the climatological position of the Aleutian Low. Based on Fig. 2, we create a straightforward precursor index (PI), which is defined by the detrended pressure difference between the Ural and the extended Aleutian area. The Ural region is

defined by the average MSLP in 45°N–80°N and 30°E–100°E, and the extended Aleutian region is defined by 45°N–80°N and 160°E–100°W. The two regions are depicted in Figure 1.

Figure 3 makes use of this precursor index and visualises the Spearman correlation coefficients between the monthly mean precursor index and the monthly mean NAM index at 10 hPa. The strong and highly significant negative correlation between the December precursor index and the January NAM at 10 hPa confirms the previous results. It indicates that a high (low)

precursor index in December favours a weak (strong) stratospheric NAM in January. The precursor index is high when there is a high-pressure anomaly in the Ural region and/or a low-pressure anomaly in the extended Aleutian area. The correlation between the December surface index and January stratospheric index shows the largest correlation coefficient of the possible combinations with a coefficient of 0.76 in ERA5 and 0.54 in ICON. However, the precursor index based on MSLP solely also works as a predictor for the stratospheric NAM in all other extended winter months from November to March, with the

only exception of February, when the negative correlation between the January precursor index and February NAM is not statistically significant in the ERA5 data, but highly significant in the ICON ensemble simulations. Overall, ICON reproduces the signals from ERA5 very well, with slightly smaller correlation coefficients but higher statistical significance due to the higher amount of realisations. However, in particular for the NAM in December and January the correlation coefficients seem to be slightly underestimated in ICON. This weaker coupling behaviour in mid-winter could be related to a climatologically

too weak stratospheric polar vortex in these ICON simulations (Köhler et al., 2021). Interestingly, the connection between the

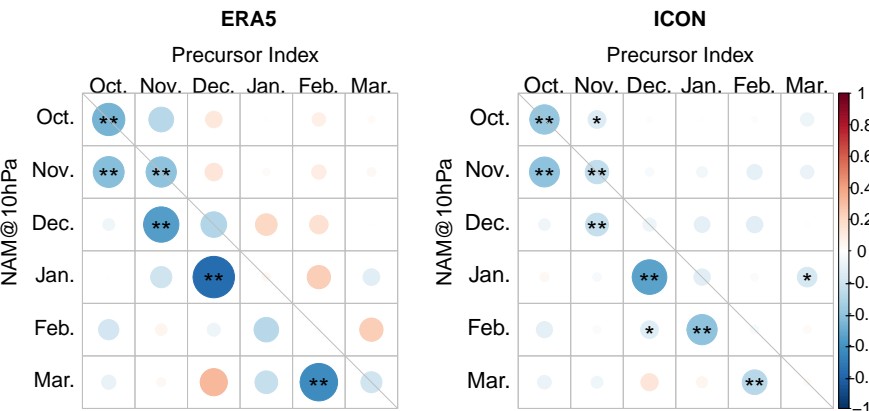

**Figure 3.** Spearman correlation between the detrended, monthly mean precursor index and NAM index at 10 hPa. Circle size and colour correspond to the strength of the correlation coefficients. Blue (red) colours denote a negative (positive) correlation and the stars denote significance, with one star indicating statistical significance at the 95% level and two stars statistical significance at the 99% level.

MSLP anomalies and the stratospheric circulation seems to be more direct in October and November, as there is already a significant negative correlation without a time lag, which disappears during late winter. The two components of the precursor index were also analysed individually and although the Ural pressure patterns contribute more strongly to the correlation with stratospheric NAM of the following month, subtracting the averaged MSLP of the extended Aleutian area gives a clear added value, in particular in late winter. Moreover, the average MSLP in the two regions does not show any statistically significant correlation between each other. Thus, indicating that the two precursor patterns evolve independently from one another. Furthermore, ENSO can be ruled out as a confounding process for the precursor index, as the Niño 3.4 index in winter is significantly correlated to the MSLP in the Aleutian region in December, but is not correlated to the Ural MSLP (not shown).

The potential of the precursor index in the context of seasonal prediction is demonstrated in Fig. 4, which shows NAM composite differences between years with a high and a low precursor index in the months November (a,b), December (c,d), January (e,f) and February (g,h) for ERA5 and ICON. The anomalies in the precursor index already tend to manifest as a tropospheric NAM signal since the precursor patterns partly project onto the tropospheric NAM. By the end of the month, the signals propagate into the stratosphere. A high (low) precursor index is clearly connected to a negative (positive) NAM signal. Although the precursor index is based solely on MSLP patterns, strong stratospheric circulation anomalies become visible in the two months following the original precursor anomaly. Moreover, these stratospheric anomalies propagate back into the troposphere leading to significant tropospheric NAM signals for up to 3 months after the original anomaly in the precursor index. Even though the basic patterns of NAM propagation are similar for the four displayed months, there are some distinct features subject to the month of the original precursor anomaly: An anomalous precursor index in November is connected to significant tropospheric NAM anomalies in the same month, but first NAM signals are already visible in October. These signals in late autumn tend to directly propagate into the stratosphere, which is in line with the results from Fig. 3.

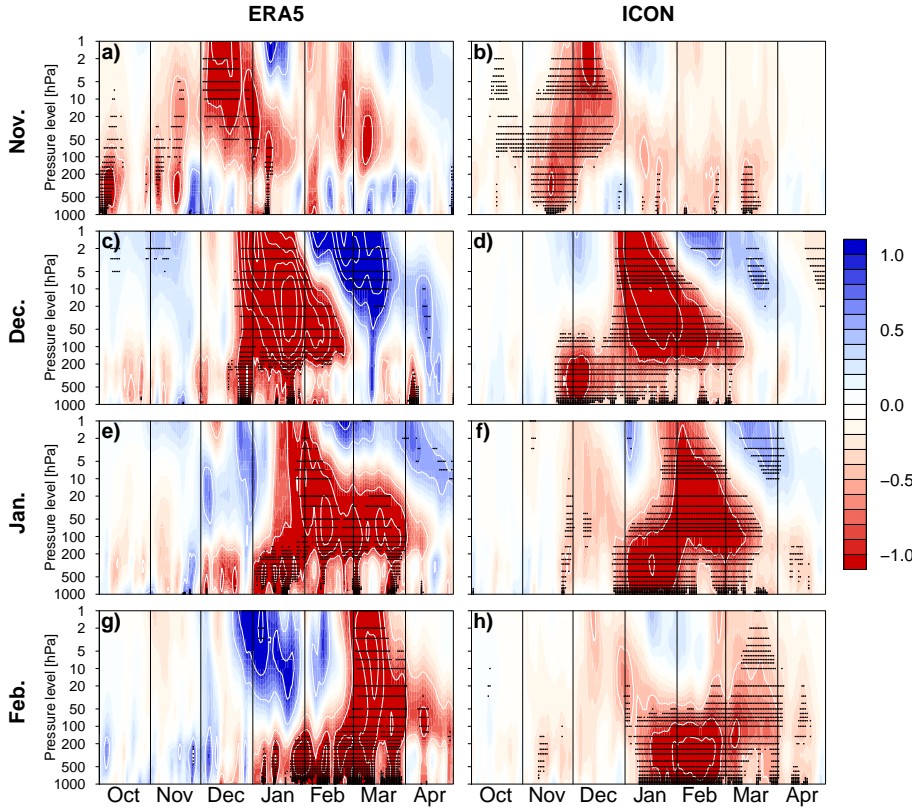

**Figure 4.** Composites of the time–height development of the NAM for the difference between high (80th percentile) and low (20th percentile) precursor index in ERA5 (a,c,e,g) and ICON (b,d,f,h). The composites are created based on the monthly mean precursor index for the months of November (a,b), December (c,d), January (e,f) and February (g,h). Each composite consists of 9 events in ERA5 and 40 events in ICON. The NAM index is nondimensional. The contour interval for the white contours is 0.5 and stippling indicates statistical significance at the 95% level according to a two-sided Wilcoxon-test. Note that due to common practice negative values (weak NAM) are red and positive values (strong NAM) are blue.

The strongest NAM signals manifest in the upper stratosphere in December, followed by a downward propagation into the lower stratosphere in January. This is also when the stratospheric signals tend to propagate back into the troposphere, thereby significantly affecting tropospheric circulation. The ICON ensemble simulations reproduce this pattern with slightly weaker intensity. However, also suggesting a more persistent influence on the tropospheric circulation also in February and March.
In December, the stratospheric response to the precursor index in ERA5 and ICON is strongest, once again in accordance with Fig. 3. The strong negative NAM signal in January and February is followed by a significant positive NAM anomaly in the upper stratosphere in February and in the lower stratosphere in spring. Precursor anomalies in January are connected to very strong tropospheric NAM signals in the same month. These propagate into the stratosphere in late January, leading to strong downward propagating signals affecting the lower stratosphere in February and March. The influence on the upper

and mid stratosphere only lasts until mid-February in ERA5, thus explaining why the correlation between the monthly mean precursor index in January and the monthly mean NAM at 10 hPa in February is weaker and not significant (cf. Fig. 3). However, the effect on the lower stratosphere is strong and highly significant. This once again impacts the troposphere yielding highly significant signals in February and March in ERA5 and ICON. A strong precursor index in February is connected to a negative tropospheric NAM in January and February. This signal then propagates into the stratosphere in late February and early March leading to a significant weakening of the stratospheric polar vortex in March and April, including the known "dripping" events into the troposphere. In general, there is no large difference in the coupling behaviour between early and late winter. Nevertheless, in the low-variability, build-up phase of the stratospheric polar vortex troposphere-stratosphere coupling is more direct with smaller time lags, whereas the coupling is less direct with larger time lags in the more variable phase of the vortex in late winter. Naturally, the strength and position of the stratospheric polar vortex will impact the amount of waves that can propagate into the stratosphere from the troposphere. Already Polvani and Waugh (2004) showed a comparable analysis using events of high and low heat flux at 100 hPa to create NAM composites. Figure 4 demonstrates that these composites can also be created by using the MSLP-based precursor index instead and thus yield similar results. Additionally, using the precursor index is connected to larger time lags in the stratosphere. The time lag between tropospheric MSLP anomalies and vertical wave fluxes will be discussed in the following section. Overall, the NAM patterns clearly demonstrate that there can be strong coupling between the troposphere and the stratosphere in NH extended winter and that a better understanding of the involved mechanisms could help to improve subseasonal to seasonal prediction. Hence, in the following chapter, we focus on these coupling mechanisms, in particular on the upward propagation of tropospheric anomalies into the stratosphere.

### 3.2 Coupling mechanisms

The MSLP precursor anomaly patterns described in the previous section are known to constructively interfere with the climatological wave-1 pattern in the troposphere (e.g. Smith and Kushner, 2012; Bao et al., 2017). To investigate how the precursor index relates to changes in the climatological wave forcing, we create composites and average the wave amplitudes over the precursor core region of 50°-70°N. In accordance with literature, a strong precursor index (80th percentile) is related to an extended winter (NDJF) average 56.6% (47%) increase of the climatological wave-1 amplitude based on the 500hPa geopotential height fields in ERA5 (ICON). Wave-1 amplitudes tend to be particularly large in mid to late winter for high precursor index states (cf. Table A1). On the other hand, a weak precursor index is related to a 19.7% (20.7%) decrease in the wave-1 amplitude. The importance of wave-1 and wave-2 forcing arises specifically because higher wave numbers cannot propagate vertically into the stratosphere in the presence of strong zonal wind (Charney and Drazin, 1961). The precursor index itself is not related to any anomalous wave-2 forcing. However, the two components of the precursor index exhibit a different behaviour. A strong Ural high-pressure system is connected to an average wave-1 and wave-2 amplitude increase of 49.6% (40.9%) and 9.7% (19.4%), respectively (cf. Table A2). And analogous, a weak Ural high corresponds to a wave-1 and wave-2 amplitude reduction. Whereas changes in the wave-1 forcing related to the Ural MSLP are stronger in mid-winter, changes in the wave-2 forcing are stronger in early winter. Surface pressure anomalies in the extended Aleutian region are also related to changes in the wave-1 amplitude with a 46.3% (32.6%) increase for a deepened Aleutian low and 31.2% (17.4%) decrease for a weak

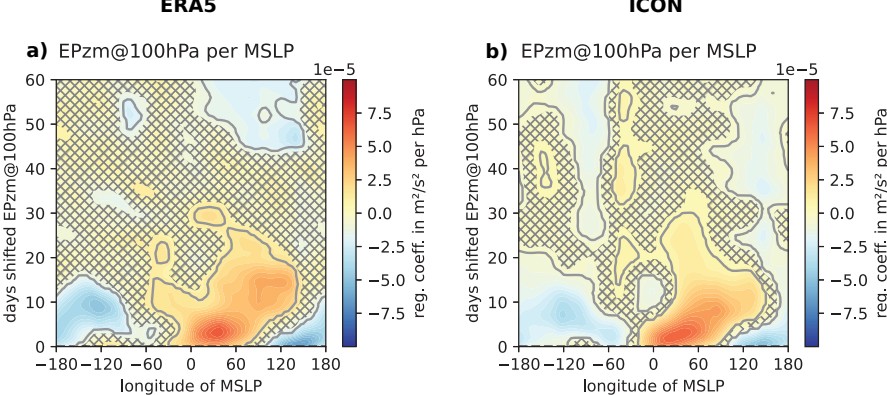

**Figure 5.** Regression of 45°N to 80°N meridional mean MSLP onto 40°N to 80°N meridional mean 100 hPa zonal mean vertical EP flux. MSLP is averaged over 10° longitude bins with their respective centre positions given along the x-axis. Daily values of MSLP from November to February of ERA5 data (a) and all years and ensemble members of ICON data (b) are used. Data of the 100 hPa zonal mean vertical EP flux is shifted by days given on the y-axis relative to MSLP data. Areas with regression coefficients with p-values above 0.05 are hatched, and areas with p-values below 0.001 are encircled.

Aleutian low (cf. Table A3). The corresponding wave-2 anomalies, however, are of the opposing sign, as a deep Aleutian low is connected to a 13.8% (1.8%) decrease of the wave-2 amplitude, and accordingly, higher pressure in the Aleutian area is linked to an increased wave-2 amplitude by 16.2% (17.0%) in ERA5 (ICON). This opposing effect of wave-1 and wave-2 amplitudes for the extended Aleutian region may explain why the Ural region is a stronger precursor for the strength of the stratospheric polar vortex than the extended Aleutian region, although the Aleutian region has an overall stronger effect on wave-1 amplitudes.

Based on planetary wave theory of Matsuno (1971), we investigate the role of tropospheric wave forcing in coupling the troposphere and stratosphere. Vertical wave propagation is quantified by means of the zonal mean Eliassen-Palm flux (Andrews and Mcintyre, 1976) as described in Sect. 2.2. As a first step, we investigate how the zonally resolved MSLP anomalies project onto the vertical component of the zonal mean EP flux during the whole winter season from November to February. The results are very similar for ERA5 in Figure 5a and ICON in Figure 5b. They show positive (negative) MSLP anomalies in the Ural region at 40°E inducing a positive (negative) vertical EP flux anomaly at 100 hPa with a time delay between 0 and 5 days of the maximum of the regression coefficients. The intensity of this positive relation weakens and reaches up to 20 days, while it also broadens regionally. The relation between negative (positive) MSLP anomalies in the Aleutian region and positive (negative) vertical EP flux anomalies is weaker. It emerges first with almost no time delay at 150°E. Then the area of strongest regression coefficients gradually moves west with a secondary maximum at 120°W with a time delay of 10 days. Both anomalies indicate that a strengthened Ural blocking and a deeper Aleutian low contribute to enhanced upward EP flux emerging in the lower stratosphere. Corresponding correlation coefficients in the Ural region are 0.27 for ERA5 and 0.25 for ICON. In the Aleutian

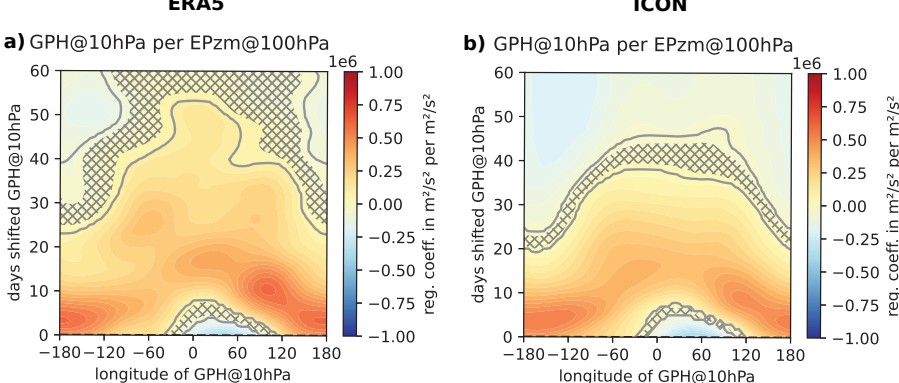

**Figure 6.** Regression of 40°N to 80°N meridional mean 100 hPa zonal mean vertical EP flux onto 45°N to 80°N meridional mean 10 hPa geopotential. Geopotential is averaged over 10° longitude bins with their respective centre positions given along the x-axis. Daily values of vertical EP flux from November to February of ERA5 data (a) and all years and ensemble members of ICON data (b) are used. Data of the 10 hPa geopotential is shifted by days given on the y-axis relative to vertical EP flux data. Areas with regression coefficients with p-values above 0.05 are hatched, and areas with p-values below 0.001 are encircled.

region, they are -0.19 in ERA5 and -0.15 in ICON. Although these values are low in terms of explained statistical variance, the regression coefficients indicate a significant contribution to upward EP flux from a typical 10 hPa MSLP anomaly, that reaches $0.001\,\mathrm{m}^2/\mathrm{s}^2$, which is about one-third of typical mean EP flux or half of the typical standard deviation.

But how does the EP flux relate to the strength of the stratospheric polar vortex by means of the zonally resolved geopotential at 10 hPa? In ERA5 (Figure 6a) and ICON (Figure 6b) we see a dominating positive relationship indicating positive (negative) EP flux anomalies are related to positive (negative) geopotential anomalies. The maximum of the signal appears first around 180°W with a time delay of 0 to 5 days. This maximum then extends westwards towards 100°E where it reaches a time delay of about 10 days relative to the initial EP flux anomaly. In particular, in ICON we further see the signal also extending eastwards.

Generally, this influence on the polar vortex shows that a weakening related to upward EP flux appears first over the Pacific and then develops into a full polar vortex weakening in the following 10 days. This can also be interpreted as a vortex displacement from the Pacific towards the Eurasian continent, which is a very typical behaviour. The positive correlation reaches 0.41 in ERA5 and 0.43 in ICON, which is again a relatively low explained statistical variance. Nevertheless, the regressions indicate that a $0.001\,\mathrm{m}^2/\mathrm{s}^2$ vertical EP flux anomaly can induce a shift of geopotential up to $1000\,\mathrm{m}^2/\mathrm{s}^2$, which is about one-fifth of

the standard deviation of geopotential at 10 hPa. The negative regressions in Figure 6 indicate the displaced more stable vortex in case of upward EP flux around 50°E in the very beginning. Furthermore, they indicate that the vortex re-emerges about a month after the initial disturbance.

Whereas the well-established zonal mean EP fluxes can explain the link between anomalies of MSLP and the strength of the stratospheric polar vortex, it does not allow for zonally-resolved analyses of the vertical coupling processes. Therefore, we want

to introduce an exploratory method that strongly focuses on the coupling mechanisms, but is not very quantitative. However,

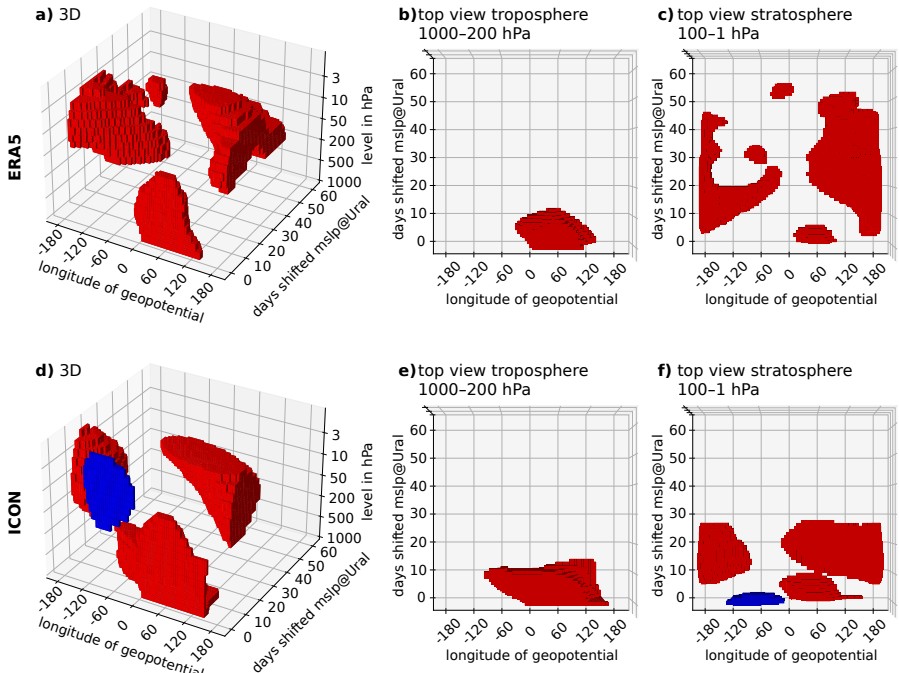

**Figure 7.** Correlation of the daily mean Ural (45°N–80°N and 30°E–100°E) MSLP in NDJF onto the daily, meridional mean geopotential (60°N–85°N) for ERA5 data (a-c) and the ICON ensemble (d-f). Figures a) and d) are a function of longitude [°], time [days], and height [hPa], whereas Figures b) and e) show a top view of tropospheric signals (≥200hPa) and Figures c) and f) show a top view of stratospheric signals (≤100hPa & ≥1hPa). Only the 10% largest absolute correlation coefficients are shown, with red (blue) colours denoting a positive (negative) correlation.

it allows for disentangling the different mechanisms of action involved in coupling tropospheric signals to the stratosphere without losing the involved timescales. Therefore, we correlate the geopotential as a function of longitude, height and time with our precursor MSLP centres of action. Figure 7 illustrates the 10% largest absolute correlation coefficients for the Ural region in November–February (NDJF). For better interpretation, we show three different views of the same data — a three-290 dimensional view (left), a top view of tropospheric signals (≥200hPa, middle), and a top view of stratospheric signals (≤100hPa & ≥1hPa, right). The tropospheric response to MSLP changes in the Ural region is mainly barotropic, i.e. high surface pressure in the Ural area is connected to higher geopotential heights above it and vice versa. This signal reaches the tropopause without a time lag but is disconnected from the main stratospheric signals, which emerge east of the Ural region and propagate west- and eastward throughout time. Thus, a high-pressure anomaly in the Ural area is related to positive stratospheric geopotential 295 anomalies that are a clear indication of a weakened stratospheric polar vortex. The strong positive correlation coefficients remain in the stratosphere for up to 55 days in ERA5 and 30 days in the ICON ensemble. We identify the EP flux (cf. Fig. 5 & 6) as a crucial link between the tropospheric instantaneous reaction at 60°E and the stratospheric response that originates at 170°E. Except for the smaller time lag, the ICON ensemble is in good agreement with the ERA5 occurrence of strong

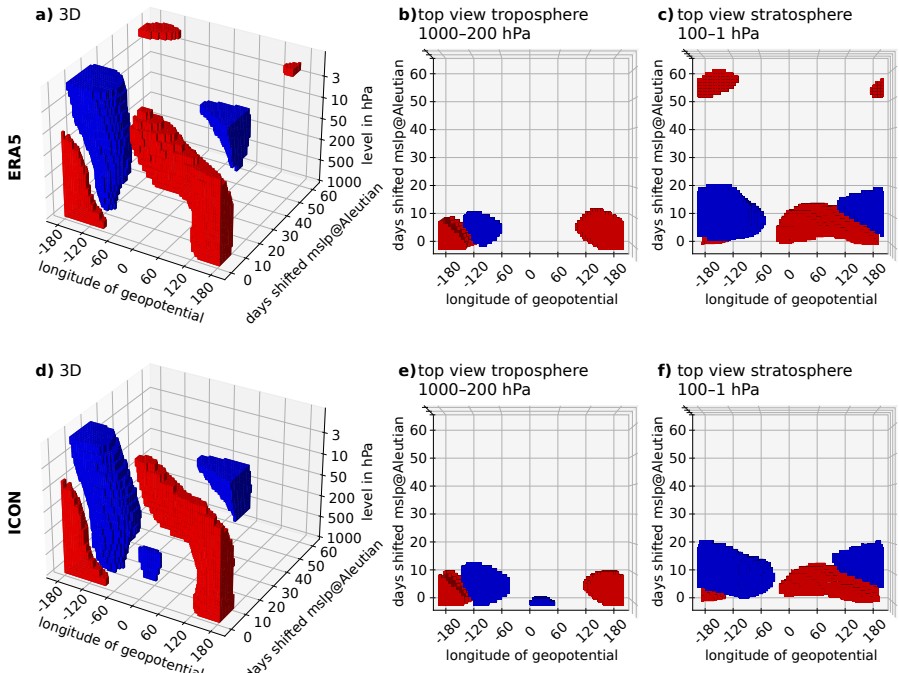

**Figure 8.** As in figure 7 but for the daily mean MSLP in the extended Aleutian area (45°N–80°N and 160°E–100°W).

regression coefficients in time and space. The negative correlation visible in ICON centered at approximately 100°W is a
remnant of the fact that the stratospheric polar vortex tends to be stronger before it breaks down / weakens. We also see this in
the ERA5 data when we go backwards in time.

    The Aleutian MSLP is also positively correlated to the geopotential above it in the troposphere (cf. Figure 8). However, it is
additionally connected to a signal of the opposite sign that emerges at approximately 500 hPa and 80°W. This pattern resembles
the two main centres of action of the PNA. It is this opposite signal over North America which has a strong vertical structure
and directly penetrates into the stratosphere. Hence, e.g. a low-pressure anomaly in the extended Aleutian low area is connected
to positive geopotential anomalies in the mid-troposphere over North America, which directly penetrate into the stratosphere
leading to a strengthening of the stratospheric Aleutian High. This stratospheric pattern is strongly linked to a displacement
of the stratospheric polar vortex towards Eurasia and the subsequent weakening of the NAM. A positive MSLP anomaly on
the other hand would be connected to a more pole-centered, strong stratospheric polar vortex. The displacement character
of the stratospheric response is reinforced by the opposite signal response towards Eurasia. The displacement character is in
agreement with the strong wave-1 amplitude increase related to a deepened Aleutian low. However, a deepened Aleutian low
is additionally connected to increased wave-3 amplitudes (cf. Table A3). As the Aleutian low is known to be one of the centres
of action of the PNA (e.g. Wallace and Gutzler, 1981), we suggest that anomalous wave-1 activity is directly induced via
the PNA. As indicated by the westward tilt of the vertical extent of the two pressure centres, vertically propagating planetary
waves potentially play a role for the influence of the Aleutian region on the stratosphere as well. Still, the relation is weaker

in terms of vertical EP flux compared to the Ural region (cf. Fig. 5), indicating differences in the specific mechanisms. The strong response in the stratospheric correlation coefficients to the Aleutian MSLP is less long-lasting than the Ural response — with up to 20 days in ERA5 and ICON. This is in agreement with the overall weaker link between the extended Aleutian MSLP and stratospheric NAM compared to the Ural MSLP. Overall, the reanalysis and model strongly agree on the processes of troposphere–stratosphere coupling for the extended Aleutian region.

## 4 Conclusions and discussion

This study conducts a comprehensive investigation of tropospheric precursors of stratospheric winter circulation by means of the NAM. Whereas part 3.1 of the results mainly focuses on the precursor patterns and involved timescales, part 3.2 investigates the involved coupling mechanisms. Based on the analysis of ERA5 reanalysis data and ICON ensemble simulations, the results from part 3.1 can be summarised as follows:

- We identify two main tropospheric precursor regions for stratospheric circulation in winter: the Ural area (45°N–80°N and 30°E–100°E) and the extended Aleutian area (45°N–80°N and 160°E–100°W). Whereas the Ural MSLP exhibits significant negative regression coefficients, the extended Aleutian MSLP exhibits significant positive regression coefficients. I.e., a weak NAM is often preceded by high-pressure anomalies in the Ural area or low-pressure anomalies in the extended Aleutian area and vice versa.

- The timescales of these precursor patterns range from 5 to nearly 60 days, which strongly matches the timescales of subseasonal to seasonal prediction. The largest stratospheric predictability arises 25–45 prior in the Ural area and 20–30 days in the extended Aleutian area.

- A simple monthly mean precursor index based on the detrended MSLP differences between the Ural and the extended Aleutian area is strongly correlated to the NAM at 10 hPa of the following month. The strongest correlation is found between the December precursor index and January NAM at 10 hPa.

- Time–height development plots of the NAM for precursor composites additionally take into account the downward coupling from the stratosphere to the troposphere. The original anomaly in the precursor index is connected to significant surface influence in the following one to three months. Thus, once again highlighting the importance of coupling processes between the troposphere and stratosphere for seasonal prediction in mid-latitude winter.

The precursor patterns in our study match findings from earlier studies (e.g. Baldwin et al., 2021; Cohen and Jones, 2011; Garfinkel et al., 2010). Additionally, we were able to identify the involved timescales by not averaging over fixed lag times. Based on these precursor patterns, in Sect. 3.2 we investigated the involved coupling mechanisms. Although MSLP anomalies in both regions constructively interfere with the wave-1 field we identified two different pathways for troposphere–stratosphere coupling:

- MSLP anomalies in the Ural area have a barotropic signal within the troposphere and are connected to increased climatological wave-1 and wave-2 amplitudes. Vertical planetary wave propagation connects these tropospheric anomalies in the Ural area to the strength of the stratospheric polar vortex. The vertical component of the EP flux is particularly sensitive to MSLP changes from 0° to 120°E.

- The pathway of coupling for the extended Aleutian region is less straightforward: MSLP anomalies in the Aleutian area are related to a pattern that resembles the two main centres of action of the PNA over the North Pacific and North America. Whereas the Aleutian pressure anomaly remains in the troposphere mainly, it triggers an opposite pressure pattern in the mid-troposphere over North America, which constructively interferes with the climatological wave-1 field and is directly induced via the PNA. This opposite signal reaches the stratosphere towards the Aleutian area. Hence, an intensified Aleutian Low in the troposphere is related to a strengthened Aleutian High in the stratosphere, which is connected to a displaced and weakened stratospheric polar vortex.

Overall, the results of this study highlight the potential of stratospheric processes for subseasonal to seasonal prediction and are in good agreement with the study from Garfinkel et al. (2010), which focuses on mid-tropospheric precursor patterns. In accordance with Domeisen et al. (2020b) we demonstrate, that the timescales of upward troposphere–stratosphere coupling can exceed the deterministic range of weather prediction. However, in this study, we investigate the involved coupling mechanisms in a climatological sense, and the described precursor patterns, pathways and timescales can differ for each event. I.e. an analysis of single events shows that 20–25% of the strong/weak stratospheric vortex events are preceded by a precursor index that is not expected to trigger the given stratospheric anomaly. Thus, it is very important to mention that the bottom-up perspective based on Matsuno (1971) cannot explain the full variability of the stratospheric polar vortex, as also the current state and position of the stratospheric polar vortex plays a large role. E.g. Birner and Albers (2017) argue that approximately only one-third of sudden stratospheric vortex decelerations are preceded by extreme anomalous upward planetary wave fluxes. Our study also shows that the zonally-averaged planetary wave approach cannot explain all stratospheric variability, as some of it arises from within the stratosphere. Additionally, we demonstrate that zonally-resolved structures in the geopotential are able to vertically penetrate into the stratosphere and push the vortex out of stability, a process that is particularly important for the Aleutian precursor pattern. Overall, this study emphasises the importance of the non-zonally-averaged picture for an in-depth understanding of troposphere–stratosphere coupling mechanisms. By using a time and space-resolved method this study offers a novel approach to investigating pathways of troposphere–stratosphere coupling.

Moreover, this is the first study to show that upward troposphere–stratosphere coupling is realistically represented in ICON. Although the overall patterns of coupling are very realistically reproduced by ICON, it tends to underestimate the coupling strength in mid-winter, which coincides with a too weak stratospheric polar vortex in these simulations (Köhler et al., 2021). In agreement with Köhler et al. (2021), we also demonstrate that the downward propagation of stratospheric anomalies back into the troposphere is realistically simulated in ICON. However, before the ICON model can make use of stratospheric processes for actual seasonal predictions, issues within the tropical stratosphere, in particular with the quasi-biennial oscillation (Köhler

et al., 2021), need to be solved. Nevertheless, the realistic representation of extratropical troposphere–stratosphere coupling in
ICON is a first step towards using the atmosphere component of ICON for seasonal predictions in future.

*Code and data availability.* ERA5 data are available on the Copernicus Climate Change Service (C3S) Climate Data Store: https://cds.
climate.copernicus.eu/cdsapp#!/dataset/reanalysis-era5-single-levels?tab=form (Hersbach et al., 2020). The ICON model code is distributed
under an institutional license issued by the Deutscher Wetterdienst (DWD). Further information can be found via https://code.mpimet.mpg.
de/projects/iconpublic. The output of the ICON simulations used in this study is made available at https://www.wdc-climate.de/ui/entry?
acronym=DKRZ_LTA_238_ds00004.

## Appendix A:  Analysis of how the precursor index and its components is linked to changes in climatological wave amplitudes

**Table A1.** Wave amplitude difference in percent for months with a high (80th percentile) / low (20th percentile) precursor index with respect to the climatology over all years for ERA5 and the ICON ensemble simulations. The wave amplitudes are calculated from the monthly mean 500 hPa geopotential height fields and the wave amplitudes of the different wave numbers (k=1-3) are averaged over the precursor core region of 50°-70°N.

| | | ERA5 | | | ICON | | |
|---|---|---|---|---|---|---|---|
| | **Wave number k** | **1** | **2** | **3** | **1** | **2** | **3** |
| **High** | **November** | 34.60% | 20.40% | -14.20% | 30.00% | 21.90% | -10.50% |
| **precursor index** | **December** | 50.90% | 0.60% | -10.00% | 47.80% | 8.50% | -15.20% |
| **(80th percentile)** | **January** | 67.80% | -11.00% | -11.70% | 57.70% | 6.60% | -33.70% |
| | **February** | 74.30% | -5.40% | -11.80% | 52.90% | 3.40% | -7.70% |
| | **NDJF mean** | 56.6% | 1.1% | -11.9% | 47.0% | 10.1% | -17.0% |
| **Low** | **November** | -35.70% | -5.10% | 18.80% | -18.80% | -31.90% | 18.50% |
| **precursor index** | **December** | -26.10% | 14.40% | 29.90% | -35.00% | 1.10% | 20.20% |
| **(20th percentile)** | **January** | -22.10% | 3.90% | 26.60% | -23.30% | -4.40% | 18.10% |
| | **February** | 6.60% | 13.70% | -3.50% | -4.30% | -2.80% | 13.80% |
| | **NDJF mean** | -19.7% | 6.6% | 18.4% | -20.7% | -9.4% | 17.7% |

**Table A2.** Same as in Table A1 but for the Ural region component of the precursor index.

| | | ERA5 | | | ICON | | |
|---|---|---|---|---|---|---|---|
| | **Wave number k** | **1** | **2** | **3** | **1** | **2** | **3** |
| **High pressure** | **November** | 19.80% | 26.50% | -48.90% | 16.10% | 30.00% | -34.90% |
| **in Ural region** | **December** | 50.30% | 12.20% | -18.60% | 34.70% | 16.50% | -38.00% |
| **(80th percentile)** | **January** | 59.70% | 0.90% | -37.20% | 57.80% | 15.70% | -36.50% |
| | **February** | 49.60% | -1.10% | -29.10% | 40.90% | 15.60% | -33.30% |
| | **NDJF mean** | 44.9% | 9.7% | -33.4% | 37.4% | 19.4% | -35.7% |
| **Low pressure** | **November** | 3.00% | -31.30% | 88.90% | -9.40% | -42.40% | 27.40% |
| **in Ural region** | **December** | -4.00% | -5.70% | 46.90% | -22.90% | -13.80% | 43.30% |
| **(20th percentile)** | **January** | -4.80% | -13.50% | 39.80% | -23.40% | -19.20% | 43.60% |
| | **February** | 1.30% | 1.10% | 23.30% | 3.30% | -27.80% | 36.50% |
| | **NDJF mean** | -1.2% | -12.5% | 49.9% | -13.5% | -25.6% | 37.7% |

**Table A3.** Same as in Table A1 but for the extended Aleutian region component of the precursor index.

| | | ERA5 | | | ICON | | |
|---|---|---|---|---|---|---|---|
| | Wave number k | 1 | 2 | 3 | 1 | 2 | 3 |
| **High pressure** | **November** | -36.90% | 1.10% | 77.70% | -21.60% | 13.10% | 1.80% |
| **in Aleutian region** | **December** | -37.80% | 14.60% | 2.90% | -30.70% | 15.10% | -25.90% |
| **(80th percentile)** | **January** | -26.50% | 21.30% | -3.20% | -12.10% | 14.40% | -30.40% |
| | **February** | -23.20% | 28.70% | -23.30% | -4.30% | 26.30% | -26.10% |
| | **NDJF mean** | -31.2% | 16.2% | 13.8% | -17.4% | 17.0% | -20.2% |
| **Low pressure** | **November** | 23.50% | -4.90% | 20.50% | 25.90% | 11.10% | 17.50% |
| **in Aleutian region** | **December** | 46.30% | -10.40% | 39.90% | 28.70% | -1.30% | 13.40% |
| **(20th percentile)** | **January** | 54.00% | -14.30% | 15.30% | 36.20% | -5.20% | 30.20% |
| | **February** | 62.30% | -26.40% | 25.80% | 40.40% | -12.30% | 13.30% |
| | **NDJF mean** | 46.3% | -13.8% | 25.4% | 32.6% | -1.8% | 18.7% |

*Author contributions.* RK and RJ contributed equally to this work. RK performed the model experiments with ICON. RJ and RK analysed the data. RJ developed the time-space varied regression analysis. RK wrote the manuscript draft. RJ and DH reviewed and edited the manuscript.

*Competing interests.* The authors declare that they have no conflict of interest.

*Acknowledgements.* The authors thank Sabine Erxleben for her technical support in calculating the Northern Annular Mode (NAM). Additionally, we thank the European Centre for Medium-Range Weather Forecasts (ECMWF) for the ERA5 data. R. Köhler is supported by the European Union's Horizon 2020 research and innovation framework programme under grant agreement no. 101003590 (PolarRES). R. Jaiser is supported by "Synoptic Events during MOSAiC and their Forecast Reliability in the Troposphere-Stratosphere System" (SynopSys)
funded by the German Federal Ministry of Education and Research (Grant/Award Number: 03F0872A). D. Handorf is partly supported by the German Research Foundation (DFG, Deutsche Forschungsgemeinschaft) Transregional Collaborative Research Center SFB/TRR 172 "Arctic Amplification: Climate Relevant Atmospheric and Surface Processes, and Feedback Mechanisms (AC)3" (Project-ID 268020496) and by the European Union's Horizon 2020 research and innovation framework programme under Grant agreement no. 101003590 (PolarRES). The authors want to acknowledge the Deutsches Klimarechenzentrum (DKRZ) in Hamburg and Alfred-Wegener-Institute Helmholtz-Zentrum
für Polar- und Meeresforschung (AWI) high performance computing team for providing the technical infrastructure to perform model runs and their analysis. We further acknowledge support by the Open Access Publication Funds of AWI.

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
