# Peer review of "How do different pathways connect the stratospheric polar vortex to its tropospheric precursors?"

_Weather and Climate Dynamics, 2023_

## Referee Comment (RC2)

**Review: How do different pathways connect the stratospheric polar vortex to its tropospheric precursors?**

Raphael H. Köhler, Ralf Jaiser, and Dörthe Handorf

April 3, 2023

**Summary**

This analysis identified and examined the pathways of troposphere-stratosphere coupling at the seasonal timescale in ERA5 and ICON model. The results highlight tropospheric precursors of stratospheric extended winter circulation and the timescales of their influence. The authors make use of the NAM index to identify the coupling mechanisms and regions: the Ural area and the extended Aleutian area. Furthermore, the authors develop a precursor index based on the MSLP difference between these two regions and show that it is mostly correlated with the subsequent NAM in the stratosphere, which highlights the importance of coupling. Moreover, vertical Plumb fluxes are investigated to identify pathways of coupling. In the Ural area vertical planetary waves propagation transmits signal from the troposphere to the stratosphere, whereas for the Aleutian region the pathway seems to be different. The geopotential anomalies in the upper layers seem to play a greater role.

The research is timely and novel as it discusses the mechanisms of the stratosphere-troposphere coupling at the subseasonal-to-seasonal timescales which can be an important source of predictability. However, as the authors mention, there is great variability across individual events. Overall, the manuscript presents results in a well-constructed way. I have some major and minor comments presented below and I recommend this manuscript for publication after addressing the comments.

**Major comments**

- As the Plumb flux is formulated using a quasi-geostrophic assumption, I would suggest using the wave activity flux by Takaya and Nakamura (2001). This flux is defined for the case of a zonally varying basic flow and focuses on the wave activity associated with planetary waves, as the wavy anomalies are considered to be embedded in the basic flow that includes the climatological planetary waves. The basic state in the Northern Hemisphere in winter shows inhomogeneities that can modulate the propagation of planetary waves.

- Regression coefficients in Figs. 2 and 6 have very different magnitudes and provide limited information as for the meaningful *effects* of this regression. While the results are meaningful for the non-hatched areas, it would be good to provide the effect-size by adding $R^2$, which shows the explained variance ratio in the dependent variable.

- In part of the analysis (Figs 4 and 5) you show the difference between high (80th percentile) and low (20th percentile) precursor index. I do not fully understand the purpose of showing the difference and not high/low index separately. It seems to me that this would be more clear for interpretation. Please, provide more explanation of your choice or maybe show the results separately for high and low index.

**Minor comments**

L77: The length of the model simulations is a bit shorter than the length of the ERA5 period chosen for the analysis. I think it would be more straightforward to take the same periods of time for the intercomparison reasons.

L112-113: Why did you choose to limit the showed correlation coefficients to the 10% largest? Might this result in the loss of information?

Fig.1 c, d: The font of the longitude labels and colorbar seems too small to me.

L129-130: Would it be better to use ensemble means for the intercomparison reasons?

Fig.2: It seems that "@100hPa" at the x-axis label is wrong.

L161-162: Please, consider adding that these areas are also shown in Fig.1.

L171: It seems interesting that "the negative correlation between January precursor index and February NAM is not statistically significant in the ERA5 data, but highly significant in the ICON ensemble simulations." Do you have any explanation for this?

L175: "The two components of the precursor index were also analysed individually": I guess this is not shown?

L179: Please, provide the sample sizes of composites. Are the same samples used also in Section 3.2?

L218: Did you test your results using the wave activity flux at different heights? Why did you choose to use 100hPa in the study?

L255-258: "This signal reaches the tropopause without a time lag but is disconnected from the main stratospheric signals, which emerge east of the Ural region and propagate west- and eastward throughout time. Thus, a high-pressure anomaly in the Ural area is connected to positive stratospheric geopotential anomalies that are a clear indication of a weakened stratospheric polar vortex." This explanation seems confusing to me, as in the first part you mention that the signal is disconnected, but then the anomalies are connected at different heights. I suggest the authors to rewrite this part more in a more clear way.

L261-262: The ICON ensemble is somewhat in agreement, but it also have strong negative correlations, how would you explain this? At least, this is worth mentioning in text.

Fig. 7 and 8: Please consider adding briefly heights to the b,c,e,f captions

---

## Author Response (AR1)

Dear editor and reviewers,

Thank you very much for the detailed comments on the original manuscript and for considering publication after revision of the manuscript. Please find attached the answers to the comments of all three reviewers on our initial manuscript with the title "How do different pathways connect the stratospheric polar vortex to its tropospheric precursors?". For your convenience we cited the reviewers' comments in separate sections, and responded in bold font and indent. The line specifications in our responses refer to the track-changes file.
Additionally, at the end of this document, we included a detailed description adapted analysis of vertical wave propagation. This was conducted based on the reviewers' comments.

We believe that the extensive changes have led to a strongly improved manuscript due to the constructive comments of the reviewers.

**Reviewer 1**

The manuscript presents a quantitative and mechanistic analysis of mean-sea-level-pressure precursor patterns observed before sudden stratospheric warmings, comparing results between ERA5 and ICON model simulations. One important contribution of the study is therefore to demonstrate the potential of ICON for studying stratosphere-troposphere coupling and for seasonal forecasting in general. It is noteworthy that the results indicate anomalous surface weather to be present for up to three months after the precursor patterns first occur. Additionally, the relatively large sample size obtained from the model simulations allows the authors to resolve the results for different months, indicating the need for careful interpretation when composites are based on events from an entire winter season.

However, parts of the mechanistic explanations were unconvincing to me in the current form. A discussion of how the applied localized wave activity analyses relate to planetary wave diagnostics would be beneficial (please see my comments below).

Overall, I found the paper well-written and interesting to read. Therefore, I recommend that the paper be published once the major concerns are addressed.

**Major comment 1:**

You have demonstrated that a strong positive difference of MSLP over the Ural region and the Pacific region, which is defined as the "precursor index", is followed by "significant tropospheric NAM signals for up to 3 months after the original anomaly in the precursor index" (ll. 184f). I agree that this is remarkable and that it can likely represent an important source of predictability for seasonal forecasting.

However, it would be beneficial to address or discuss the potential role of any confounding processes. Specifically, it would be worthwhile to explore whether ENSO teleconnections could project on both the identified precursor regions, particularly over the Pacific, and anomalies of the polar vortex.

> **We want to thank Reviewer 1 for this comment, as we agree that it is important to discuss potentially confounding processes in the manuscript. Different studies have connected El Niño with a deepened Aleutian low and a ridge downstream over North America (e.g.,**

O'Reilly, 2018; Soulard et al., 2019). To further investigate how ENSO relates to anomalies in the precursor regions and of the stratospheric polar vortex, we calculated the ENSO3.4 index and correlated it to the Precursor index, MSLP in the Aleutian and Ural area as well as the NAM at 10hPa. Please find our results for the ERA5 reanalysis in Figure R1. The only significant correlation between the indices is found for ENSO3.4 and the Aleutian region, where El Niño is connected to a deepened Aleutian low in December. The Ural region does not exhibit any significant correlation to the ENSO index, so that we don't assume that ENSO is a confounding process. The non-significant correlation between ENSO and NAM at 10hPa in January seems to modulated via the Aleutian low, i.e., El Ni Niño is connected to a deepened Aleutian low in December, which then favours a weaker stratospheric polar vortex in January. A further confounding factor could be global warming, but by detrending the data before calculating the correlations we can exclude this factor. We included a short discussion of this analyses into the revised manuscript (cf. ll. 203-206).

O'Reilly, C.H. Interdecadal variability of the ENSO teleconnection to the wintertime North Pacific. *Clim Dyn* 51, 3333–3350 (2018). https://doi.org/10.1007/s00382-018-4081-y

Soulard, N., Lin, H. & Yu, B. The changing relationship between ENSO and its extratropical response patterns. *Sci Rep* 9, 6507 (2019). https://doi.org/10.1038/s41598-019-42922-3

**Major comment 2:**

*I have some concerns about the Plumb flux analysis in the second part of the study. It would be helpful if the authors provided more information on the formulation used for the vertical Plumb flux and referenced previous successful applications.*

**We are happy to provide more information about the Plumb flux. Nevertheless, based on the comments of the other reviewers, we changed the manuscript so that it is no longer used. From the perspective of Eurasian surface anomalies and their influence on stratosphere-troposphere coupling, Cohen et al. 2007 was potentially one of the first influential applications of the vertical component of the Plumb 1985 flux. We take the freedom to cite from there:**

**The vertical component of the WAF is given by**

$$ F = \Omega p \sin(2\phi) S^{-1} \{ \upsilon' T' - [2\Omega a \sin(2\phi)]^{-1} \partial(T'\Phi')/\partial\lambda \}, $$

**where Ω is the angular velocity of rotation, p is pressure, φ is latitude, υ is meridional velocity, T is temperature, a is the mean radius of the earth, Φ is geopotential, λ is longitude, and primes indicate the departure from the local climatological mean. The static stability S = ⟨–(p/H)∂T/∂p + κT/H⟩, where angle brackets indicate the 20°–90°N polar-cap and climatological average, κ = R/cp and H is the scale height. The field F is useful for localizing in longitude and latitude the source of vertically propagating stationary planetary Rossby waves. The zonal mean of F is proportional to the meridional eddy heat flux and the vertical component of the quasigeostrophic Eliassen–Palm flux.**

Cohen, J., Barlow, M., Kushner, P. J., & Saito, K. (2007). Stratosphere–troposphere coupling and links with Eurasian land surface variability. Journal of Climate, 20(21), 5335-5343.

Our formulation is basically the same, while we further describe the basic state to calculate the perturbations (primed quantities) in the manuscript: We use a 91-day rolling mean annual cycle of the full ERA5 data or full ICON ensemble, respectively. We consider only low-frequency or large-scale contributions to the Plumb Flux and therefore apply a low-pass filter sensitive to timescales above 10 days to the perturbations.

Additionally, we provide a list of a few very recent studies where Plumb 1985 flux has been applied in the context of troposphere-stratosphere coupling, showing that it is widely and actively used by reputable authors in the field:

Finke, Kathrin, Hannachi, Abdel, and Hirooka, Toshihiko, 2023, "Exceptionally persistent Eurasian cold events and their stratospheric link" Asia-Pacific Journal of Atmospheric Sciences 1976-7951

Ding, Xiuyuan, Chen, Gang, Zhang, Pengfei, Domeisen, Daniela I. V., and Orbe, Clara, 2023, "Extreme stratospheric wave activity as harbingers of cold events over North America" Communications Earth & Environment Vol. 4, No. 1, 2662-4435

Gastineau, Guillaume, Frankignoul, Claude, Gao, Yongqi, Liang, Yu-Chiao, Kwon, Young-Oh, Cherchi, Annalisa, Ghosh, Rohit, Manzini, Elisa, Matei, Daniela, Mecking, Jennifer, Suo, Lingling, Tian, Tian, Yang, Shuting, and Zhang, Ying, 2023, "Forcing and impact of the Northern Hemisphere continental snow cover in 1979–2014" The Cryosphere Vol. 17, No. 5, pp 2157, 1994-0424

Zhang, J., Orsolini, Y.J., Limpasuvan, V. et al. Impact of the Pacific sector sea ice loss on the sudden stratospheric warming characteristics. npj Clim Atmos Sci 5, 74 (2022). https://doi.org/10.1038/s41612-022-00296-w

Unlike the Eliassen-Palm flux (divergence), I am unaware of how localized Plumb flux theoretically relates to changes in the zonal mean flow, which could be discussed more in detail (e.g., see ll. 234f: "The connection between wave activity in 100 hPa and the stratospheric polar vortex is not as straightforward as one might expect, as increased Plumb flux does not necessarily imply a weakening of the stratospheric polar vortex.").

In our understanding, Plumb 1985 flux (as well as the flux defined by Takaya and Nakamura (2001)) is a generalization of the original Eliassen-Palm relation with the aim to be zonally resolved. These zonally resolved quantities describe a more generalized form of wave activity and therefore do not have a direct relation to the zonal wind via their divergence. Nevertheless, authors of these studies describe that these kinds of wave activity can be related to westerly momentum that adds to a zonal mean basic flow. Thus, additional wave activity can have an influence on the westerly flow (in our case the stratospheric polar vortex). We intended to build on that to find and/or explain physical pathways of our precursor index.

Nevertheless, this comment and comments from other reviewers motivated our switch to conventional zonal mean EP fluxes in the revised version of the manuscript. These give a clear formal link between changes of zonal mean zonal wind and additional wave activity.

I wonder if the presented precursor geopotential anomalies in Figure 1 might reflect anomalous planetary wave activity, given their projection on the climatological wave-1. An intensified wave-2 pattern could possibly explain why anomalies are larger over the Ural region than over the Pacific, potentially questioning if it were in fact two distinct mechanisms that relate the observed precursor signals over the Ural and over the Pacific to the stratospheric anomalies.

> **Thank you very much for this comment. Based on it, we calculated the wave amplitudes related to our Precursor index. As expected, we see that the Precursor index strongly projects onto the climatological wave-1, with increased wave-1 amplitudes during high Precursor index states and decreased wave-1 amplitudes during low index states (cf. Table 1 and 2). The overall wave-2 amplitude changes are small for the Precursor index. However, as you suspected wave-2 amplitude differences could explain, why the Ural region might affect the stratosphere stronger than the Aleutian region, although the Aleutian region is connected to larger wave-1 amplitude changes (cf. Table 3,4 and 5,6). Whereas a strong (weak) Ural high is linked to increased (decreased) wave-1 and wave-2 activity (Table 3 & 4), a low (high) Aleutian low is connected to increased (decreased) wave-1 amplitudes, but decreased (increased) wave-2 amplitudes (Table 5 & 6). Hence, for MSLP anomalies in the extended Aleutian region wave-2 can oppose the sign of wave-1 activity. This can explain why climatologically the Ural region is the main tropospheric driver of the stratospheric polar vortex.**
> **As we think it is very beneficial for the understanding of the involved mechanisms, we also included this discussion into our revised manuscript (cf. ll. 245-267).**

In addition, I have two suggestions regarding the Plumb flux analyses. First, providing typical orders of magnitude for the Plumb flux and its variability in Figure 5 would be helpful in interpreting the observed anomalies of up to 0.02 m^2/s^2. Second, providing typical correlation coefficients associated with the plotted regression coefficients in Fig. 6 may aid in interpreting the amount of co-variability.

> **Thank you for pointing out this issue. Related to comments from other reviewers we decided not to show Plumb fluxes in the revised manuscript. It was replaced by zonal mean Eliassen-Palm flux diagnostics as defined by the Transformed Eulerian Mean equations. A discussion can be found at the end of the response document. Nevertheless, we provide the requested information for other plots where applicable in the updated manuscript (e.g., ll. 296-299 or 314-316).**

**Minor comments:**

- ll. 39-41: maybe you could be a bit more clear about what you mean by "not predictable beyond deterministic timescales"; (e.g., in the provided reference, Karpechko 2018 concludes that "[...] days 8–12 before the events, the SSWs can be considered predictable in a probabilistic sense. After day 7 the predictability can be considered to be close to deterministic.")

> **This statement is based on Domeisen et al. (2020b). To make things clearer, we reformulated the sentence to "However, they also demonstrate that stratospheric extreme events themselves tend to exhibit similar predictability to tropospheric weather and, in particular, SSW events tend to be less predictable." (cf. ll. 41-43)**

- ll. 77: Have you tested using only winters where you have both ICON simulations and ERA5 data available (1979/80 to 2016/17)? Alternatively, you may want to add a small comment why you think this is not necessary.

**We performed some basic test using ERA5 data only up to winter 2016/2017 and received the same qualitative results. Due to the very large variability of the stratospheric polar vortex in winter, we decided to maximise our sample size by including the whole ERA5 period available at this point in time.**

- ll 86-89: You may want to consider stating which spatial domain was used to calculate the EOFs, if the data was area-weighted for the calculation of the EOFs and how ERA5 and ICON data was deseasonalized.

  **We improved the description of the calculation of the NAM indices by adding information about the area, the removing of the seasonal cycle and the area weighting. The text of former ll. 86-89 was extended accordingly (cf. ll. 95-100):**

    **The basic mean of quantifying the strength of the stratospheric polar vortex in this study is the Northern annular mode (NAM), which is the dominant pattern of dynamic variability in the extratropical Northern Hemisphere (NH). Based on Baldwin and Thompson (2009), we used an empirical orthogonal function (EOF) analysis of the daily, zonally-averaged geopotential anomalies at each pressure level over the area 20ºN to 90ºN to calculate the NAM. Geopotential height anomalies were obtained by removing the mean seasonal cycle at each grid point, which is calculated for each data set separately (ERA5 and ICON) by averaging the respective geopotential heights day by day over all years. To ensure equal-area weighting, the data was weighted by the square root of the cosine of latitude before performing the EOF analysis. The standardized corresponding timeseries of the first principal components are the NAM indices.**

- Fig. 1: I'm unsure about what you mean by "Composites of [...] MSLP [...] for the same January weak vortex events in ERA5 and ICON". If I'm correct then events in ERA5 and ICON are basically independent (sorry for my ignorance if not)?

  **Thanks for pointing out this misleading wording. "The same" refers to the same as were used to generate Figure 1 a and b. The events were calculated separately for ERA5 and ICON. We added "as in (a) and (b)" to the caption of Figure 1.**

- Fig. 2: The presented composites are based on a strongly negative January-mean NAM. Given the relatively long timescales in the stratosphere, I would imagine that in some cases, the negative NAM in January results from a SSW that happened already in December. In that case, part of the surface signal observed in December could in principle correspond already to a surface response of the SSW. In case you agree, you could consider adding a comment. However, I see that it is not strictly necessary as you will address the time lag between tropospheric precursor and stratospheric anomaly more in detail in Fig. 2.

  **We assume that the comment relates to Figure 1. We agree that SSW events in late December could potentially impact the NAM in January. However, we would like to point out, that the downward propagation of stratospheric signals into the troposphere usually happens one to two months after the stratospheric extreme event. Therefore, we think it is highly unlikely that the precursor patterns are generated by weak vortex events in December. Moreover, the surface response to SSWs strongly differs from the precursor patterns visible in Figure 1 c) and d) (cf. Fig. 1a in Afargan-Gerstman, H., & Domeisen, D. I., 2020).**

**Afargan-Gerstman, H., & Domeisen, D. I. (2020). Pacific modulation of the North Atlantic storm track response to sudden stratospheric warming events. *Geophysical Research Letters*, *47*(2), e2019GL085007.**

- ll. 130: A two-sample significance test could provide additional confirmation that the observed differences between ERA5 and ICON are explainable by sampling.

  **When we look into the single ICON ensemble members, we see that signals are also noisier in ICON. Ensemble approaches usually make the overall signal smoother due to the larger sample size. Therefore, we note that a qualitative agreement between ICON and ERA5 is given. Accordingly, we adjusted the original statement to: "The larger sample size in ICON results in a smoother image, while the qualitative agreement between ICON and ERA5 is given." (cf. l. 149).**

- Fig. 2: I expect "@ 100hPa" to be wrong.

  **Yes, this is a mistake and was corrected in the revised manuscript.**

- Fig. 2: Is it surprising that ICON shows significant regression coefficients even after 60 days and longer? (How much longer?)

  **We agree with the reviewer's comment - this is surprising. Nevertheless, the regression coefficients become gradually weaker indicating its actual very low influence. The statistically significant relation ends at about 80 days. Furthermore, we note that this particular relation appears only in December. Other months are more similar to the results of ERA5.**

- Fig. 4: How many events are shown in each composite?

  **For ERA5 there are 9 events for each month and for the ICON ensemble 40 events each. We added this information to the caption of Figure 4.**

- ll. 181-182: Is this because the precursor pattern directly projects onto the NAM or is it because the precursor pattern connects with a polar vortex anomaly which then leads to an altered NAM index?

  **This is a good remark, thank you. In our opinion, the precursor pattern partly projects onto the tropospheric NAM. This is particularly the case for the Aleutian region (cf. Fig. 1c and e.g. https://www.cpc.ncep.noaa.gov/products/precip/CWlink/daily_ao_index/ao.loading.shtml). Hence, a deepened Aleutian low is connected to a negative NAM in the troposphere and vice versa. Moreover, the Ural region partly overlaps with the low-pressure centre of the AO. Thus, high pressure in the Ural region also favours a negative NAM and vice versa. Overall, this seems to suggest that the precursor pattern directly projects onto the tropospheric NAM, with a high (low) precursor index favouring a negative (positive) NAM. We added this information into the revised manuscript (cf. ll. 211-212).**

- Fig. 6: Maybe you want to add "at 100 hPa" after meridional mean Plumb flux.

  **That is a good idea, thank you. However, Fig. 6 has been replaced.**

- Fig. 7: Only the 10% largest absolute correlation coefficients are shown. What is the obtained threshold for the correlation coefficient?

> **The threshold values in terms of the 10th percentile of absolute correlation coefficients are listed referring to the respective figures:**
>
> **Figure 7a (ERA5, Ural): 0.19**
>
> **Figure 7b (ICON, Ural): 0.12**
>
> **Figure 8a (ERA5, Aleutian): 0.16**
>
> **Figure 8b (ICON, Aleutian): 0.14**
>
> **These correlation coefficients are low, which is related to the high amount of noise introduced from daily values. Even the highest correlation between slp and geopotential at 1000 hPa in the origin region of either Ural or Aleutian reach only 0.82 … 0.89.**

- l. 287: "and/or": this might be beyond the scope of your study, but I imagine it would be indeed interesting to understand whether the combined pattern emerges from the composite averaging or really occurs combined during individual events

  > **As our two regions do not show any correlation, we assume that the the pattern emerges from composite averaging. Yet, when we look at single events in ERA5, there can be some events where we see both, a Ural and an Aleutian precursor. This motivated us to write "and/or". As the patterns are, however, uncorrelated we decided to write only "or" in the revised manuscript (cf. l. 375).**

- l. 307: "Vertical planetary wave fluxes do not explain the link between extended Aleutian MSLP and the stratospheric NAM. Here it is a pattern that resembles the two main centers of action of the PNA which directly penetrates into the stratosphere.": Fig. 8c does show a clear displacement, i.e., a wave 1 signal in the stratosphere and to my understanding, Fig. 8b does not exclude the possibility that anomalous wave 1 activity is also present in the troposphere, even if strongly induced via the PNA. It would be useful if you could clarify/ discuss this when considering my major comment 2.

  > **As discussed in the response to major comment 2, anomalies in MSLP in the Aleutian region are strongly related to anomalous wave-1 activity in the troposphere. However, a deepened Aleutian low is additionally connected to increased wave-3 activity (cf. Table 5). Hence, we suggest that anomalous wave-1 activity is induced via the PNA. We included this into the revised manuscript (e.g., ll. 358-361).**

- General comment: To my understanding, (quasi-)observed SSTs are prescribed in the simulations. To what extent do you think does this need to be considered when interpreting the good agreement between ERA5 and ICON results?

  > **Yes, the SSTs and sea ice cover are prescribed using CMIP6 standard protocol for AMIP simulations. More specifically, it is based on a merged observational product of the UK MetOffice HadISST and NCEP OI2. Although SSTs and sea ice are prescribed in ERA5 and ICON, providing similar diabatic heating forcing for Rossby waves, one cannot expect that the propagation of Rossby waves and the coupling of stratosphere and troposphere are per se simulated realistically in ICON, because too many atmospheric processes are involved. Additionally, it is standard practice to investigate stratospheric circulation using AMIP-style simulations.**

[Figure]

**Figure R1.** Spearman correlation between the detrended, monthly mean ENSO3.4 index and the Precursor index (a), the NAM at 10 hPa (b), the Ural index (c) and the Aleutian index (d) in ERA5. Circle size and colour correspond to the strength of the correlation coefficients. Blue (red) colours denote a negative (positive) correlation and the stars denote significance, with one star indicating statistical significance at the 95% level and two stars statistical significance at the 99% level.

| | ERA5 | | | | ICON | | | |
|---|---|---|---|---|---|---|---|---|
| k | 1 | 2 | 3 | 4 | 1 | 2 | 3 | 4 |
| **Nov** | 34.60% | 20.40% | -14.20% | -23.80% | 30.00% | 21.90% | -10.50% | 48.40% |
| **Dec** | 50.90% | 0.60% | -10.00% | 9.20% | 47.80% | 8.50% | -15.20% | 46.70% |
| **Jan** | 67.80% | -11.00% | -11.70% | 57.40% | 57.70% | 6.60% | -33.70% | 59.80% |
| **Feb** | 74.30% | -5.40% | -11.80% | 117.70% | 52.90% | 3.40% | -7.70% | 54.80% |
| **NDJF** | 56.6% | 1.1% | -11.9% | 38.8% | 47.0% | 10.1% | -17.0% | 52.3% |

**Table R1.** Wave amplitude difference in percent between months with a high Precursor index (80[th] percentile) and the climatology over all years for ERA5 and ICON ensemble simulations. The wave amplitudes are calculated from the monthly mean 500hPa geopotential height fields and the wave amplitudes of the different wave numbers (1-4) are averaged over the precursor core region of 50°-70°N.

| | ERA5 | | | | ICON | | | |
|---|---|---|---|---|---|---|---|---|
| k | 1 | 2 | 3 | 4 | 1 | 2 | 3 | 4 |
| **Nov** | -35.70% | -5.10% | 18.80% | 31.20% | -18.80% | -31.90% | 18.50% | 20.30% |
| **Dec** | -26.10% | 14.40% | 29.90% | -19.10% | -35.00% | 1.10% | 20.20% | -29.50% |
| **Jan** | -22.10% | 3.90% | 26.60% | -22.80% | -23.30% | -4.40% | 18.10% | 11.50% |
| **Feb** | 6.60% | 13.70% | -3.50% | -29.60% | -4.30% | -2.80% | 13.80% | -8.90% |
| **NDJF** | -19.7% | 6.7% | 18.4% | -10.0% | -20.7% | -9.4% | 17.7% | -1.7% |

**Table R2.** Same as Table 1 but for low Precursor index (20[th] percentile) vs. climatology

| | ERA5 | | | | ICON | | | |
|---|---|---|---|---|---|---|---|---|
| k | 1 | 2 | 3 | 4 | 1 | 2 | 3 | 4 |
| **Nov** | 19.80% | 26.50% | -48.90% | 11.50% | 16.10% | 30.00% | -34.90% | 44.50% |
| **Dec** | 50.30% | 12.20% | -18.60% | 29.80% | 34.70% | 16.50% | -38.00% | 15.10% |
| **Jan** | 59.70% | 0.90% | -37.20% | 17.60% | 57.80% | 15.70% | -36.50% | 34.70% |
| **Feb** | 49.60% | -1.10% | -29.10% | 77.20% | 40.90% | 15.60% | -33.30% | 45.80% |
| **NDJF** | 44.9% | 9.7% | -33.4% | 33.2% | 37.4% | 19.4% | -35.7% | 34.6% |

**Table R3.** Same as Table 1 but for high Ural index (80[th] percentile) vs. climatology

| | ERA5 | | | | ICON | | | |
|---|---|---|---|---|---|---|---|---|
| k | 1 | 2 | 3 | 4 | 1 | 2 | 3 | 4 |
| **Nov** | 3.00% | -31.30% | 88.90% | 64.40% | -9.40% | -42.40% | 27.40% | 24.70% |
| **Dec** | -4.00% | -5.70% | 46.90% | 5.00% | -22.90% | -13.80% | 43.30% | 14.20% |
| **Jan** | -4.80% | -13.50% | 39.80% | 17.80% | -23.40% | -19.20% | 43.60% | 21.70% |
| **Feb** | 1.30% | 1.10% | 23.30% | -20.50% | 3.30% | -27.80% | 36.50% | 17.10% |
| **NDJF** | -1.2% | -12.5% | 49.9% | 17.1% | -13.5% | -25.6% | 37.7% | 19.4% |

**Table R4.** Same as Table 1 but for low Ural index (20[th] percentile) vs. climatology

| | ERA5 | | | | ICON | | | |
|---|---|---|---|---|---|---|---|---|
| k | 1 | 2 | 3 | 4 | 1 | 2 | 3 | 4 |
| **Nov** | 23.50% | -4.90% | 20.50% | 5.70% | 25.90% | 11.10% | 17.50% | 25.80% |
| **Dec** | 46.30% | -10.40% | 39.90% | -15.50% | 28.70% | -1.30% | 13.40% | 41.70% |
| **Jan** | 54.00% | -14.30% | 15.30% | 34.40% | 36.20% | -5.20% | 30.20% | 46.90% |
| **Feb** | 62.30% | -26.40% | 25.80% | 86.30% | 40.40% | -12.30% | 13.30% | 38.20% |
| **NDJF** | 46.3% | -13.8% | 25.4% | 26.5% | 32.6% | -1.8% | 18.7% | 38.2% |

**Table R5.** Same as Table 1 but for low Aleutian index (20th percentile) vs. climatology

| | ERA5 | | | | ICON | | | |
|---|---|---|---|---|---|---|---|---|
| k | 1 | 2 | 3 | 4 | 1 | 2 | 3 | 4 |
| **Nov** | -36.90% | 1.10% | 77.70% | 43.40% | -21.60% | 13.10% | 1.80% | -40.10% |
| **Dec** | -37.80% | 14.60% | 2.90% | 46.90% | -30.70% | 15.10% | -25.90% | -56.90% |
| **Jan** | -26.50% | 21.30% | -3.20% | -33.50% | -12.10% | 14.40% | -30.40% | -34.50% |
| **Feb** | -23.20% | 28.70% | -23.30% | -38.20% | -4.30% | 26.30% | -26.10% | -24.50% |
| **NDJF** | -31.2% | 16.2% | 13.8% | 5.3% | -17.4% | 17.0% | -20.2% | -39.2% |

**Table R6.** Same as Table 1 but for high Aleutian index (80th percentile) vs. climatology

**Reviewer 2**

**Summary**

This analysis identified and examined the pathways of troposphere-stratosphere coupling at the seasonal timescale in ERA5 and ICON model. The results highlight tropospheric precursors of stratospheric extended winter circulation and the timescales of their influence. The authors make use of the NAM index to identify the coupling mechanisms and regions: the Ural area and the extended Aleutian area. Furthermore, the authors develop a precursor index based on the MSLP difference between these two regions and show that it is mostly correlated with the subsequent NAM in the stratosphere, which highlights the importance of coupling. Moreover, vertical Plumb fluxes are investigated to identify pathways of coupling. In the Ural area vertical planetary waves propagation transmits signal from the troposphere to the stratosphere, whereas for the Aleutian region the pathway seems to be different. The geopotential anomalies in the upper layers seem to play a greater role.

The research is timely and novel as it discusses the mechanisms of the stratosphere-troposphere coupling at the subseasonal-to-seasonal timescales which can be an important source of predictability. However, as the authors mention, there is great variability across individual events. Overall, the manuscript presents results in a well-constructed way. I have some major and minor comments presented below and I recommend this manuscript for publication after addressing the comments.

**Major comments**

- As the Plumb flux is formulated using a quasi-geostrophic assumption, I would suggest using the wave activity flux by Takaya and Nakamura (2001). This flux is defined for the case of a zonally varying basic flow and focuses on the wave activity associated with planetary waves, as the wavy anomalies are considered to be embedded in the basic flow that includes the climatological planetary waves. The basic state in the Northern Hemisphere in winter shows inhomogeneities that can modulate the propagation of planetary waves.

  **We acknowledge the comment by the reviewer and notice a similar comment by reviewer 3. We agree, it makes sense to implement the more advanced version of a 3D wave activity flux by Takaya and Nakamura (2001), although they still use quasi-geostrophic (QG) approximations. As listed in the responses to reviewer 1, Plumb fluxes are actively implemented in other very recent studies, while references of implementations of the vertical component of Takaya Nakamura fluxes are barely seen in literature.**

  **More persuasive, we recognize the argument of difficulties in the definition of the basic flow. In our case, we used a basic flow defined by a 91-day rolling mean seasonal cycle derived from the mean of all available years. Varying this definition leads to different results. This means, we agree that inhomogeneities of the winter climatology can modulate the propagation of planetary waves and these inhomogeneities cannot be addressed by a climatological mean state which covers multiple decades. Regarding this, we want to cite from the original Takaya and Nakamura study:**

  > **The wave-activity flux is a useful diagnostic tool for illustrating a "snapshot" of a propagating packet of stationary or migratory QG wave disturbances and thereby for inferring where the packet is emitted and absorbed, as verified in several applications to the data.**

**Regarding the reviewer's comments, the cited passage from the original study and our actually much worse results when we implement the Takaya Nakamura flux, we now believe a substantial redesign of our approach would be necessary to implement such a diagnostic in a climatological way since it is only useful on "snapshots". In our climatological study, we need a general definition of a basic flow that is not based on snapshots. Therefore, we decided to reject the use of 3D wave activity flux and implemented similar diagnostics with zonal mean Eliassen-Palm flux based on Transformed Eulerian Mean equations, which is described at the end of this response document.**

**The actual implementation of Takaya Nakamura fluxes in the same fashion as in Figure 6 led to degraded signals (cf. Figure R2), which we explain with the aforementioned discussion.**

- Regression coefficients in Figs. 2 and 6 have very different magnitudes and provide limited information as for the meaningful effects of this regression. While the results are meaningful for the non-hatched areas, it would be good to provide the effect-size by adding $R^2$, which shows the explained variance ratio in the dependent variable.

    **We provide the requested metrics for corresponding figures. Please recognize that Figure 6 has changed due to the first comment. In Figure 2a (ERA5) the maximum value of the correlation coefficient is 0.26, the minimum value -0.35. In Figure 2b (ICON) the maximum value of the correlation coefficient is 0.18, the minimum value -0.24. Indeed, these values are very low, which is related to noise introduced by the evaluation of daily data. Nevertheless, we point out that the regression indicates that a reasonable 10hPa sea level pressure anomaly can push the NAM index by up to 0.5 index points, which is a significant change of the circulation. Furthermore, we present further evidence of the predictive skill of our precursor index in Figures 3 and 4. While the statistical explained variance is low, we still find skill in the given relation. The manuscript was adapted accordingly (e.g., ll. 174-177, 296-299, 314-316)**

- In part of the analysis (Figs 4 and 5) you show the difference between high (80th percentile) and low (20th percentile) precursor index. I do not fully understand the purpose of showing the difference and not high/low index separately. It seems to me that this would be more clear for interpretation. Please, provide more explanation of your choice or maybe show the results separately for high and low index.

    **We chose to show the differences between the high and the low precursor index, as the signals of the individual differences with respect to the climatology are rather similar with an opposite sign. By only showing the differences between the two extreme states of the Precursor index we reduce the number of figures without a significant loss of information. To confirm this, we attach the same figure for the individual extreme state differences with respect to the climatology (Figure R3 for high Precursor index minus climatology and Figure R4 for low Precursor index minus climatology). Figure 5 of the manuscript was replaced by a different Figure, which does not use the composite approach.**

**Minor comments**

L77: The length of the model simulations is a bit shorter than the length of the ERA5 period chosen for the analysis. I think it would be more straightforward to take the same periods of time for the intercomparison reasons.

**We did some basic testing by using ERA5 data only up to the winter 2016/2017 and received the same qualitative results. Due to the very large variability of the stratospheric polar vortex in winter, we, however, decided to maximise our sample size by including the whole ERA5 period available at this point in time.**

L112-113: Why did you choose to limit the showed correlation coefficients to the 10% largest? Might this result in the loss of information?

**We tested the approach with several thresholds. If the threshold is increased, either areas with already visible results become simply larger, or additional noise is introduced in particular at longer time shifts. The given threshold is a compromise to arrive at a clear image with distinguishable anomalies.**

Fig.1 c, d: The font of the longitude labels and colorbar seems too small to me.

**Thank you for pointing this out. It was adjusted in the revised manuscript.**

L129-130: Would it be better to use ensemble means for the intercomparison reasons?

**When we look into the single ICON ensemble members, we see that signals are also noisier in ICON compared to the ensemble mean and thus more similar to ERA5. Ensemble approaches usually make the overall signal smoother due to the larger sample size. The ensemble approach thereby minimises the internal variability and maximises the actual signal. Moreover, using ensemble means as input data for the process-oriented analyses will interfere with the analyses as the daily development will differ due to the internal variability. Therefore, we first do the process-oriented analyses before averaging over the ensemble member. Overall, we note that a qualitative agreement between ICON and ERA5 is given. Accordingly, we adjusted the original statement to: "The larger sample size in ICON results in a smoother image, while the qualitative agreement between ICON and ERA5 is given." (cf. l. 149).**

Fig.2: It seems that "@100hPa" at the x-axis label is wrong.

**Thanks for pointing this out. This was corrected in the revised manuscript.**

L161-162: Please, consider adding that these areas are also shown in Fig.1.

**That is a good idea, thank you. It was added (cf. ll. 183-184).**

L171: It seems interesting that "the negative correlation between January precursor index and February NAM is not statistically significant in the ERA5 data, but highly significant in the ICON ensemble simulations." Do you have any explanation for this?

**This is a good point. An explanation can be given with the help of Figure 4e. The NAM composite for the January Precursor extremes demonstrates that there is a strong significant link between the Precursor index in January and the strength of the stratospheric polar vortex in the following two months (February and March). However, in the upper stratosphere the signal is more short-lived. Figure 3 is based on the NAM at 10hPa and Figure 4e exemplifies that the negative NAM signals remain at 10hPa only up to mid-February, whereas the signal is long-lived in the lower stratosphere. Therefore, we assume that this non-significance could be related to the limited sample size in ERA5.**

L175: "The two components of the precursor index were also analysed individually": I guess this is not shown?

**Yes, this is not shown to reduce the overall number of Figures. However, as we describe in the manuscript the Ural component contributes more strongly to the correlation with the NAM at 10 hPa, yet the Aleutian component adds significant value. The 1-month lag correlation coefficients increase by 34% for O-N, 19% for D-J, 70% for J-F and 30% for F-M, yet decrease by 22% for N-D when comparing Ural index to the combined Precursor index. Figure 2 also exemplifies that both regions are linked to strength of the stratospheric polar vortex.**

L179: Please, provide the sample sizes of composites. Are the same samples used also in Section 3.2?

**For ERA5 the sample size is 9 for each extreme event and for ICON it is 40. To make things clearer we also add this information into the manuscript (cf. caption of Figure 4). The same samples were used in Figure 5 of Section 3.2. This Figure was, however, replaced by a Figure using a different method.**

L218: Did you test your results using the wave activity flux at different heights? Why did you choose to use 100hPa in the study?

**We assess from literature that 100 hPa is a standard height to diagnose wave activity flux that reached the stratosphere. It is important to use a height level, which is safely above the tropopause. In our experience, the wave activity flux becomes relatively uniform above the tropopause, if the deviations are not too big. At lower altitudes, in particular when reaching tropospheric levels, signals become very noisy.**

L255-258: "This signal reaches the tropopause without a time lag but is disconnected from the main stratospheric signals, which emerge east of the Ural region and propagate west- and eastward throughout time. Thus, a high-pressure anomaly in the Ural area is connected to positive stratospheric geopotential anomalies that are a clear indication of a weakened stratospheric polar vortex." This explanation seems confusing to me, as in the first part you mention that the signal is disconnected, but then the anomalies are connected at different heights. I suggest the authors to rewrite this part more in a more clear way.

**Thanks for pointing this out. We agree that the wording is somewhat confusing. Replacing "... in the Ural area is connected to positive stratospheric geopotential ..." with "… in the Ural area is related to positive stratospheric geopotential…" should make it clearer (cf. l. 339).**

L261-262: The ICON ensemble is somewhat in agreement, but it also have strong negative correlations, how would you explain this? At least, this is worth mentioning in text.

**This negative correlation is a remnant of the fact that the stratospheric polar vortex tends to be stronger before it breaks down / weakens. We also see this in the ERA5 data when we go backwards in time. This information was added to the revised manuscript (cf. ll. 344-346).**

Fig. 7 and 8: Please consider adding briefly heights to the b,c,e,f captions

**This was added in the revised figures.**

[Figure]

**Figure R2.** Regression 100 hPa Takaya Nakamura flux onto the 10 hPa NAM. The figure is produced in the same way as figure 6 in the original manuscript.

[Figure]

**Figure R3.** Composites of the time–height development of the NAM for the difference between high precursor index (80th percentile) and the climatology in ERA5 (a,c,e,g) and ICON (b,d,f,h). The composites are created based on the monthly mean precursor index for the months November (a,b), December (c,d), January (e,f) and February (g,h). The NAM index is nondimensional. The contour interval for the white contours is 0.5 and stippling indicates statistical significance at the 95% level according to a two-sided Wilcoxon-test. Note that due to common practice negative values (weak NAM) are red and positive values (strong NAM) are blue.

[Figure]

**Figure R4.** Composites of the time–height development of the NAM for the difference between low precursor index (20th percentile) and the climatology in ERA5 (a,c,e,g) and ICON (b,d,f,h). The composites are created based on the monthly mean precursor index for the months November (a,b), December (c,d), January (e,f) and February (g,h). The NAM index is nondimensional. The contour interval for the white contours is 0.5 and stippling indicates statistical significance at the 95% level according to a two-sided Wilcoxon-test. Note that due to common practice negative values (weak NAM) are red and positive values (strong NAM) are blue.

**Reviewer 3**

This paper revisits the role of a strengthened Ural high and Aleutian low for variability of the NH polar vortex. In agreement with previous work, the authors find that a Ural high and an Aleutian low leads to a weaker vortex. The more novel aspects of this paper are: 1) inclusion of ICON results; 2) a focus on daily timescales to better isolate the relevant timescales; 3) a discussion of which calendar month the response peaks. Finally, the authors include a discussion of mechanisms, but as discussed below I find this discussion to be limited and problematic. Major revisions are needed before I can give a final assessment of the quality of this paper/

**Major comments:**

1. As discussed above, I don't find the mechanism part of this paper convincing. The authors use Plumb 1985 fluxes which were derived explicitly for stationary waves, and apply this method for time varying waves! There are better fluxes available (Plumb 1986, or even better Takaya and Nakamura 2001) that are more appropriate for the application. Hence any conclusions reached using a unsuitable formulation should be treated with caution.

> **We acknowledge the comment by the reviewer and notice a similar comment by reviewer 2. We agree, it makes sense to implement the more advanced version of a 3D wave activity flux by Takaya and Nakamura (2001). As listed in the responses to reviewer 1, Plumb fluxes are actively implemented in other very recent studies, while references of implementation of the vertical component of Takaya Nakamura fluxes are barely seen in literature.**
>
> **The actual implementation of Takaya Nakamura fluxes in the same fashion as in the original Figure 6 led to degraded signals. The corresponding Figure R2 can be found in the response to reviewer 2. In part, we explain reduced signals by difficulties in defining an appropriate climatological basic state. In our case, we used a basic flow defined by a 91-day rolling mean seasonal cycle derived from the mean of all available years. Varying this definition leads to different results. To discuss this further, we want to cite from the original Takaya and Nakamura study:**
>
> > **The wave-activity flux is a useful diagnostic tool for illustrating a "snapshot" of a propagating packet of stationary or migratory QG wave disturbances and thereby for inferring where the packet is emitted and absorbed, as verified in several applications to the data.**
>
> **Taking this into account, we believe a substantial redesign of our approach would be needed to implement such a diagnostic in a climatological way since it is only useful on "snapshots". In our climatological study, we need a general definition of a basic flow that is not based on snapshots. Therefore, we decided to reject the use of 3D wave activity flux and implemented a similar diagnostic with the zonal mean Eliassen-Palm flux based on the Transformed Eulerian Mean equations, which is described at the end of this response document.**

Further, they appear to ignore the previously proposed mechanism. As discussed in Garfinkel et al 2010 (already cited) and Smith and Kushner 2012 (not cited) among many others, transients that are in phase with the climatological wave-1 or wave-2 field enhance it, and via constructive interference lead to an overall increase in wave-driving. It just so happens that a transient Ural high and Aleutian low are in phase with the stationary waves, and hence enhance it. The importance of wave-1 and wave-2 arises specifically because higher wavenumbers cannot propagate vertically in the presence of

strong zonal wind (Charney and Drazin 1961). Do the authors think this mechanism is wrong? unimportant? They cite Garfinkel et al 2010 and Bao et al 2017, and so I assume they are aware of it. This mechanism was found to be important for the stratospheric polar vortex response to a range of external forcings, as discussed in detail in Smith and Kushner (see references therein) and dozens since that have cited Garfinkel et al 2010 and Smith and Kushner.

> **The authors agree that these mechanisms are important and were so far under-represented in the manuscript. Therefore, we decided to add an extensive analysis on how the precursor index and its components relate to the amplitudes of the different wave numbers based on the 500hPa geopotential in ERA5 and ICON (cf. Tables R1-6). As expected, we see that the Precursor index strongly projects onto the climatological wave-1, with increased wave-1 amplitudes during high Precursor index states and decreased wave-1 amplitudes during low index states (cf. Table R1 and R2). The overall wave-2 amplitude changes are small for the Precursor index. However, wave-2 amplitude differences could explain, why the Ural region might affect the stratosphere stronger than the Aleutian region, although the Aleutian region is connected to larger wave-1 amplitude changes (cf. Table R3,4 and R5,6). Whereas a strong (weak) Ural high is linked to increased (decreased) wave-1 and wave-2 activity (Table R3 & R4), a low (high) Aleutian low is connected to increased (decreased) wave-1 amplitudes, but decreased (increased) wave-2 amplitudes (Table R5 & R6). Hence, for MSLP anomalies in the extended Aleutian region wave-2 can oppose wave-1 activity. This could explain why the Ural region is the main tropospheric driver of the stratospheric polar vortex. This analysis is now part of the revised manuscript (cf. ll. 245-267). Moreover, we shortly discuss the basics of wave-driving in the introduction (cf. ll. 46-47), thereby including important references, such as Smith and Kushner (2012) and Charney and Drazin (1961).**

That the authors appear to ignore the previous proposed mechanism is particularly grating due to statements along the likes of "there is only a very vague understanding of the localised coupling mechanisms and involved timescales, in particular when it comes to connecting tropospheric precursor patterns to the strength of the stratospheric polar vortex. " (line 3) or "pressure anomalies in the North Pacific region, has received less attention in literature focusing on precursor patterns of SSWs" (line 53). These two statements are just plain incorrect.

> **The authors got rid of these statements in the revised manuscript (cf. ll. 2-4, 55-57). Instead, they discuss the important findings from recent literature, including e.g. Garfinkel et al. (2010) and Smith and Kushner (2012) concerning the wave-driving mechanisms (cf. ll. 46-47, 245-246) and e.g., Ineson and Scaife (2009) concerning the role of the Pacific for strength of the stratospheric polar vortex (cf. ll. 57-58).**

2. Why should the trop->strat coupling differ from early winter to late winter? Why would ICON have too weak coupling? These questions should be answered in the revised paper (assuming these differences across months and between data sources are statistically signififcant).

> **These questions are addressed in the revised manuscript in more detail (cf. ll. 235-239, 252-253, 259-260). Overall, the authors do not see any large qualitative changes in troposphere-stratosphere coupling between early and late winter (cf. Figure 3 and 4 from the manuscript). However, the strength and position of the stratospheric polar vortex will impact the amount of waves that can propagate into the stratosphere from the troposphere (Charney and Drazin, 1961). In the low-variability, build-up phase of the stratospheric polar vortex troposphere-stratosphere coupling is more direct with smaller time lags, whereas the coupling is less direct in the more variable phase of the vortex in**

late winter (cf. Figures 3 and 4 from the manuscript). Moreover, the tropospheric wave forcing related to the precursor patterns can differ between early and late winter (cf. Tables R1-6). In particular the wave-1 amplitudes tend to increase in mid to late winter for high Precursor index states.

In the revised manuscript the authors also go into more detail when comparing the ICON results to the results of ERA5 (cf. ll. 196-198, 427-428). The ensemble approach in ICON tends to smooth out strong peaks in the apparent signals, thus giving the impression that the overall signals are not as strong, although they agree qualitatively. When looking into individual ensemble members instead of the ensemble mean we see that amplitudes tend to be comparable to ERA5. However, one factor that could potentially contribute to differences in the coupling behaviour is the mean state of the stratospheric polar vortex in ICON, as it tends to be too weak in mid-winter (cf. Köhler et al., 2021). This temporally coincides with the somewhat smaller correlation coefficients in ICON in mid-winter (cf. Figure 3 of the manuscript).

Köhler, R., Handorf, D., Jaiser, R., Dethloff, K., Zängl, G., Majewski, D., & Rex, M. (2021). Improved Circulation in the Northern Hemisphere by Adjusting Gravity Wave Drag Parameterizations in Seasonal Experiments With ICON-NWP. *Earth and Space Science*, *8*(3), e2021EA001676. https://doi.org/10.1029/2021EA001676

It is not clear to this reviewer how Plumb fluxes would help answer this question. In contrast, the stationary waves in the different calendar months or different data sources will differ, and hence the heat flux anomaly for a given strengthed height or MSLP anomaly will differ (Watt-Meyer and Kushner 2015).

We appreciate the input of the reviewer. This further motivated us to use a more established metric to diagnose the vertical propagation of planetary waves. We provide an overview about our changed discussion implementing zonal mean Eliassen-Palm flux based on Transformed Eulerian Mean equations at the end of this response document.

**minor comments:**

line 38 the most recent multi-model and multi-SSW assessment of SSW predictability is by Chwat et al 2022

Thanks for pointing this out. We added this citation to the revised manuscript (cf. l. 39).

xlabel of figure 2: MSLP is not at 100hPa.

This mistake was corrected in the revised manuscript.

line 145 "conceptual" seems an odd word to choose here

To make it mor specific we replaced "conceptual" with "time-resolved" in the revised manuscript (cf. l. 164).

line 260: you would need to look at the zonal component of the Plumb flux to better relate the zonal position of anomalies in the troposphere with that in the stratosphere, though as discussed above I don't find Plumb fluxes useful. An alternate plot would be something similar to figure 2 of Ineson and Scaife 2009, but created for a range of lags.

We appreciate the idea to produce a figure that resolves wave activity fluxes in zonal and vertical direction as well as in time. This would be a similar approach as we took in our

**figures 7 and 8. Nevertheless, we removed the application 3D wave activity fluxes from our manuscript due to the several other comments from the reviewers.**

Smith, K.L. and Kushner, P.J., 2012. Linear interference and the initiation of extratropical stratosphere-troposphere interactions. *Journal of Geophysical Research: Atmospheres*, *117*(D13).

Watt-Meyer, Oliver, and Paul J. Kushner. "The role of standing waves in driving persistent anomalies of upward wave activity flux." *Journal of Climate* 28, no. 24 (2015): 9941-9954.

Chwat, Dvir, Chaim I. Garfinkel, Wen Chen, and Jian Rao. "Which Sudden Stratospheric Warming Events Are Most Predictable?." *Journal of Geophysical Research: Atmospheres* 127, no. 18 (2022): e2022JD037521.

Plumb, R.A., 1986. Three-dimensional propagation of transient quasi-geostrophic eddies and its relationship with the eddy forcing of the time—mean flow. *Journal of Atmospheric Sciences*, *43*(16), pp.1657-1678.

Plumb, R.A., 1986. Three-dimensional propagation of transient quasi-geostrophic eddies and its relationship with the eddy forcing of the time—mean flow. *Journal of Atmospheric Sciences*, *43*(16), pp.1657-1678.

Ineson, S, and A. A. Scaife. "The role of the stratosphere in the European climate response to El Niño." *Nature Geoscience* 2, no. 1 (2009): 32-36.

Takaya, K., & Nakamura, H. (2001). A formulation of a phase-independent wave-activity flux for stationary and migratory quasigeostrophic eddies on a zonally varying basic flow. *Journal of the Atmospheric Sciences*, *58*(6), 608-627.

**New zonal mean EP flux based diagnostics**

Related to the several critical points that the reviewers brought up regarding our application of Plumb 1985 fluxes, we recognized the inadequate applicability of 3D wave activity fluxes in our climatological case. In particular in the stratosphere, it is uncertain, if a zonally resolved wave flux and the attempt to connect it to a surface anomaly is reasonable. As brought up by the reviewers and shown by our newly introduced discussion of wave numbers, the surface pressure anomalies project on low wave number patterns. Therefore, it is more straight forward, if the resulting wave activity is a zonal mean measure.

We implement an analysis of conventional zonal mean Eliassen-Palm (EP) flux (Andrews and McIntyre, 1976) based on tools implemented by Jucker (2021a, b). These fluxes further give a formal relation between zonal wind and wave activity, which is not the case for 3D-based approaches. Since we now have only zonal mean information, we decided to first analyse how the zonally resolved sea level pressure (slp) anomalies project onto the vertical component of the zonal mean EP flux during the whole winter season from November to February. The results are very similar for ERA5 in Figure 5a and ICON in Figure 5b. They show positive (negative) slp anomalies in the Ural region at 40°E inducing a positive (negative) vertical EP flux anomaly in 100 hPa with a time delay between 0 and 5 days of the maximum of the regression coefficients. The intensity of this positive relation weakens and reaches up to 20 days, while it also broadens regionally. The relation between negative (positive) slp anomalies in the Aleutian region and positive (negative) vertical EP flux anomalies is smaller. It emerges first with almost no time delay at 150°E. Then the relation gradually moves west with a secondary maximum at 120°W with a time delay of 10 days. Both anomalies indicate that a strengthened Ural blocking and a deeper Aleutian low contribute to enhanced upward EP flux emerging in the lower stratosphere. Corresponding correlation coefficients in the Ural region are 0.27 for ERA5 and 0.25 for ICON. In the Aleutian region they are -0.19 in ERA5 and -0.15 in ICON. Although these values are low in terms of explained statistical variance, the regression coefficients indicate a significant contribution to upward EP flux from a typical 10 hPa slp anomaly, that reaches 0.001 $m^2/s^2$, which is about one third of typical mean EP flux or half of the typical standard deviation. We replaced Figure 5 of the original manuscript with Figure R5 and adapted the corresponding discussion.

Next, we explore how vertical EP flux anomalies in 100 hPa influence the zonally resolved geopotential field in 10 hPa depending on time during winter from November to February. In ERA5 (Figure 6a) and ICON (Figure 6b) we see a dominating positive relationship indicating positive (negative) EP flux anomalies inducing positive (negative) geopotential anomalies. The maximum of the signal appears first around 180°W with a time delay of 0 to 5 days. This maximum then extends westwards towards 100°E where it reaches a time delay of about 10 days relative to the initial EP flux anomaly. In particular in ICON we further see the signal also extending eastwards. Generally, this influence on the polar vortex shows that a weakening related to upward EP flux appears first over the Pacific and then develops into a full polar vortex weakening in the following 10 days. This can also be interpreted as a vortex displacement from the Pacific towards the Eurasian continent, which is a very typical behaviour. The positive correlation reaches 0.41 in ERA5 and 0.43 in ICON, which is again a relatively low explained statistical variance. Nevertheless, the regressions indicate that a 0.001 $m^2/s^2$ vertical EP flux anomaly can induce a shift of geopotential up to 1000 $m^2/s^2$, which is about one fifth of the standard deviation of geopotential in 10 hPa. The negative regressions in Figure 6 indicate the displaced more stable vortex in case of upward EP flux around 50°E in the very beginning. Furthermore, they indicate that the vortex re-emerges about a month after the initial disturbance. We replaced Figure 6 of the original manuscript with Figure R6 and adapt the corresponding discussion.

[Figure]

**Figure R5.** Regression of 45°N to 80°N meridional mean MSLP onto 40°N to 80°N meriodonal mean 100 hPa zonal mean vertical EP flux. MSLP is averaged over 10° longitude bins with their respective centre positions given along the x-axis. Daily values of MSLP from November to February of ERA5 data (a) and all years and ensemble members of ICON data (b) are used. Data of the 100 hPa zonal mean vertical EP flux is shifted by days given on the y-axis relative to MSLP data. Areas with regression coefficients with p-values above 0.05 are hatched, areas with p-values below 0.001 are encircled.

[Figure]

**Figure R6.** Regression of 40°N to 80°N meridional mean 100hPa zonal mean vertical EP flux onto 45°N to 80°N meridional mean 10hPa geopotential. Geopotential is averaged over 10° longitude bins with their respective centre positions given along the x-axis. Daily values of vertical EP flux from November to February of ERA5 data (a) and all years and ensemble members of ICON data (b) are used. Data of the 10 hPa geopotential is shifted by days given on the y-axis relative to vertical EP flux data. Areas with regression coefficients with p-values above 0.05 are hatched, areas with p-values below 0.001 are encircled.

Andrews, D. G., & Mcintyre, M. E. (1976). Planetary waves in horizontal and vertical shear: The generalized Eliassen-Palm relation and the mean zonal acceleration. *Journal of Atmospheric Sciences*, *33*(11), 2031-2048.

Jucker, M. (2021a). Scaling of Eliassen-Palm flux vectors. *Atmospheric Science Letters*, *22*(4), e1020.

Jucker, M., (2021b): mjucker/aostools: v2.3.2 (Version v2.3.2). – Zenodo. DOI:10.5281/zenodo.4588067

---

## Referee Report (RR1)

**Review: How do different pathways connect the stratospheric polar vortex to its tropospheric precursors?**

Raphael H. Köhler, Ralf Jaiser, and Dörthe Handorf

**Comments to the Authors**

I would like to thank the authors for careful consideration of my comments and the comments of two other reviewers. I think that the revised manuscript clarifies points where I was was not fully convinced and I appreciate the authors testing some suggestions I made. I think that using the Eliassen-Palm flux benefited the study and made the results more robust. I therefore only have minute comments/typo corrections left, after which I see the manuscript ready to be published.

**Minor comments**

L104 maybe 'EP flux **at** 100 hPa'?

L139 '..the qualitative agreement between ICON and ERA5 **is given**.' I am not sure that I understand the word 'given' in this context: given where? If you refer to some figures, please, provide the figure numbers, or, maybe, you mean 'is shown'/'can be seen (Figure xx)'/etc.?

L379 Appendix: 'amplitudes'

---

## Author Response (AR2)

**Response to Reviewers after 2nd review**

**Dear editor and reviewers,**

**Thank you very much for revising this manuscript a second time. The authors very much appreciate the comments and the positive feedback. Please find attached the answers to the comments of all three reviewers on our first revision of "How do different pathways connect the stratospheric polar vortex to its tropospheric precursors?". Although Reviewer 1 and 2 suggested to accept the manuscript as it, they included very minor comments and therefore we decided to also respond to these.**

**For your convenience we cited the reviewers' comments in separate sections and responded in bold font and indent. The line specifications in our responses refer to the track-changes file.**

**Reviewer 1**

I thank the authors for a careful revision of their manuscript.

My initial concern about a potential confounding influence of ENSO on tropospheric features and the polar vortex has been addressed. No significant linear correlation between the ENSO 3.4 index and the polar vortex was found, supporting the proposed direct upward influence of an Atleutian low/ Ural high to the polar vortex.

In the initial manuscript, the authors investigated upward stratosphere-troposphere coupling by means of the Plumb flux, which I found difficult to interpret physically. In the revised manuscript, this was addressed by substituting the Plumb flux by the Eliassen-Palm flux, which in my view significantly improves the study's mechanistic reasoning.

The authors suggest that geopotential anomalies over the Ural and Atleutian area penetrate the stratospheric polar vortex via two distinct mechanisms. Anomalies over the Ural area are linked to vertical wave propagation. Atleutian pressure anomalies near the surface are linked to anomalies in the mid-troposphere of opposite sign, which reach up into the stratosphere and displace the polar vortex.
I appreciate the authors' explanation; however, I continue to ponder the feasibility of also reconciling the latter perspective with the concept of upward-propagating waves. For example, I could imagine the negative mid-troposphere geopotential anomaly over North America to stem from the vertical westward tilt of vertically propagating Rossby waves. Near the surface, anomalies might vanish due to presence of wavenumber 3 structures, as found by the authors.

> **We find the comment of the reviewer a worthwhile contribution to the interpretation of our results and added a comment in lines 314-316.**

If that was the case, the authors' conclusion "that the zonally-averaged planetary wave approach cannot explain all stratospheric variability" (l. 359) would be somewhat misleading.

Rather, the same picture could be described from different angles.
Nevertheless, I acknowledge that this is speculative, and I genuinely appreciate the authors's efforts and explanations. All my other comments have been satisfactorily addressed. Therefore, I am happy to recommend the manuscript for publication in its current form, leaving my thoughts as potential avenues for future investigations.

**We agree that line 359 might be slightly misleading and therefore added additional explanation (cf. ll. 367-368).**

**Reviewer 2**

**Comments to the Authors**
I would like to thank the authors for careful consideration of my comments and the comments of two other reviewers. I think that the revised manuscript clarifies points where I was was not fully convinced and I appreciate the authors testing some suggestions I made. I think that using the Eliassen-Palm flux benefited the study and made the results more robust. I therefore only have minute comments/typo corrections left, after which I see the manuscript ready to be published.

**We kindly thank Reviewer 2 for the positive feedback on our revised manuscript and for pointing out some minute issues.**

**Minor comments**
L104 maybe 'EP flux at 100 hPa'?

**Thank you pointing out this mistake. We changed this throughout the manuscript.**

L139 '..the qualitative agreement between ICON and ERA5 is given.' I am not sure that I understand the word 'given' in this context: given where? If you refer to some figures, please, provide the figure numbers, or, maybe, you mean 'is shown'/'can be seen (Figure xx)'/etc.?

**We changed the word "given" to "shown" and added the reference to the figure (cf. l. 139).**

L379 Appendix: 'amplittudes'

**This was also corrected (l. 387).**

**Reviewer 3**

This is my second review of this paper. My initial criticisms were that the paper didn't adequately discuss the previous relevant literature, and that if this previous literature were taken into consideration, the present paper makes mostly incremental progress. The paper is somewhat improved on both respects, however there is still much work to be done.

Several of the key results and figures are essentially updates of Garfinkel et al 2010 (already

cited). This includes the regression whereby the Aleutian low and Ural high are both used to predict vortex strength (termed the precursor index in the current paper), and also analyzing the wavenumber-decomposed EP flux associated with anomalies in the Aleutian Low and Ural high. As best as I can tell the results are in agreement with this previous work.

There is an improvement in the current draft in the discussion section, however this reviewer still thinks that too much time and attention is devoted to repeating analyses rather than focusing on the open questions still outstanding.

> **We want to thank Reviewer 3 for the feedback on our revised manuscript. The authors agree with R3 that the results are in good agreement with Garfinkel et al. (2010). We further highlight this in the revised manuscript (cf. ll. 358-359). However, we also want to point out that our study strongly focusses on the involved timescales without using fixed lead-lag times. This allows for more precisely determining the involved time scales in days for each step in the chain from surface anomaly -> vertical wave flux anomaly -> stratospheric NAM anomaly -> downward influence on troposphere. Additionally, we investigate the involved mechanisms for individual winter months and discuss differences between early and late winter (e.g., ll. 221-225). The innovative approach in Figures 7 and 8 enables a disentanglement of the coupling mechanisms for the Ural and Aleutian region, which nicely adds to the explanation using wavenumber theory. We furthermore want to point out, that these mechanisms have so far not been analysed with the ICON model. As this unified next-generation global numerical weather prediction and climate modelling system is envisaged for seasonal predictions in coming years, we want to stress the importance of investigating these coupling mechanisms in ICON.**

**minor comments:**
line 97 climatological -> planetary [climatological amplitudes are not modified by anomalies in the Aleutian low or Ural high region]
> **Thank you for pointing this out. This was corrected (cf. l. 97).**

figure 1 implies that the Aleutian low is not significantly correlated to the vortex strength, however the rest of the figures do suggest that there is a significant relationship.
> **This is related to the different methods. Fig. 1 is an introductory figure and based on a simple composite using the monthly mean NAM@10hPa to select years with a weak vortex in January (20$^{th}$ percentile), leading to only 9 events in ERA5. This small sample size in combination with the monthly averaging is connected to the difference in significance. The more sophisticated approach of Fig. 2 uses daily mean MSLP and NAM values for a regression. Also, for the ICON ensemble (40 events) we see a clear significant signal in the Aleutian region in Figure 1. Thus, hinting that the ensemble size in ERA5 might be too small. Additionally, we would like to point out, that small regions in Fig. 1 c) are statistically significant at the 95% level, i.e., in the Gulf of Alaska and towards the Kamchatka Peninsula.**

figure 4 and accompanying discussion: Please compare to Polvani and Waugh 2004
> **Thank you for pointing out this study. We now also compare our results to the results of Polvani and Waugh (2004) (cf. ll. 225-229)**